# LLMs are Greedy Agents: Effects of RL Fine-tuning on Decision-Making Abilities

**Thomas Schmied**[1,+]  **Jörg Bornschein**[2]  **Jordi Grau-Moya**[2]
**Markus Wulfmeier**[2]  **Razvan Pascanu**[2]
[1] ELLIS Unit, LIT AI Lab, Institute for Machine Learning, JKU Linz, Austria
[2] Google DeepMind, [+] Work done at Google DeepMind

## Abstract

The success of LLMs has sparked interest in various agentic applications. A key hypothesis is that LLMs, leveraging common sense and Chain-of-Thought (CoT) reasoning, can effectively explore and efficiently solve complex domains. However, LLM agents have been found to suffer from sub-optimal exploration and the knowing-doing gap, the inability to effectively act on knowledge present in the model. In this work, we systematically study *why* LLMs perform sub-optimally in decision-making scenarios. In particular, we closely examine three prevalent failure modes: greediness, frequency bias, and the knowing-doing gap. We propose mitigation of these shortcomings by fine-tuning via Reinforcement Learning (RL) on self-generated CoT rationales. Our experiments across multi-armed bandits, contextual bandits, and Tic-tac-toe demonstrate that RL fine-tuning enhances the decision-making abilities of LLMs by increasing exploration and narrowing the knowing-doing gap. Finally, we study both classic exploration mechanisms, such as $\epsilon$-greedy, and LLM-specific approaches, such as self-correction and self-consistency, to enable more effective fine-tuning of LLMs for decision-making.

## 1 Introduction

Large Language Models (LLMs) pre-trained on massive internet-scale datasets have demonstrated success across diverse domains, including text generation and language understanding (Radford et al., 2019; Brown et al., 2020b; Team et al., 2023b; 2024a; Dubey et al., 2024). Their broad pre-training distribution enables generalization to a wide range of scenarios, including coding assistance (Li et al., 2022), education (Team et al., 2024d), and medicine (Saab et al., 2024). Therefore, their success has sparked interest in using LLMs for decision-making problems (Chen et al., 2023; Krishnamurthy et al., 2024; Nie et al., 2024) at the core of agentic AI systems (Durante et al., 2024).

One key hypothesis is that LLMs can generate informed action predictions without extensive environment interaction (Lu et al., 2024) due to "world knowledge" present in the model. Moreover, Chain-of-Thought (CoT) (Wei et al., 2022) reasoning equips models with the ability to reason about the observed history and their actions, which facilitates environment interaction. However, these advantages often do not seem to materialize into strong performance when LLMs are faced with decision-making scenarios. Notably, Krishnamurthy et al. (2024) and Nie et al. (2024) found that LLMs do not robustly engage in *exploration*, resulting in sub-optimal behavior. Similar shortcomings of LLMs have been observed by Paglieri et al. (2024) and Ruoss et al. (2024) on stateful environments commonly used in RL (e.g., grid-worlds, Atari). Both works broadly attribute the shortcomings to the *knowing-doing gap*, which states that models can possess knowledge about a task or can describe the consequences of their behavior (i.e., they know what to do), but cannot materialize this knowledge when acting (i.e., incapable of doing). Consequently, sub-optimal exploration and the knowing-doing gap are considerable obstacles towards more powerful and robust agentic LLMs.

In this work, we aim to understand *why* LLMs often perform sub-optimally in simple decision-making scenarios. In particular, we systematically study three prevalent failure modes in small-to-medium-scale LLMs across different model families: greediness, frequency bias, and the knowing-doing gap (see Section 4.2). Our analysis shows that final performance often remains sub-optimal, because LLMs prematurely commit to greedy action selection strategies, leading to stagnating action coverage

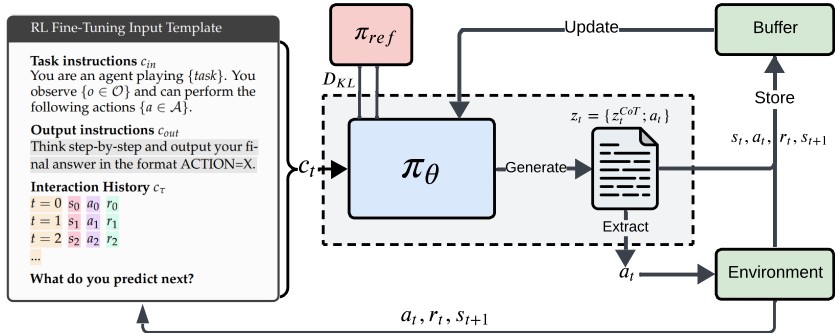

**Figure 1:** Illustration of our **Reinforcement Learning Fine Tuning (RLFT)** pipeline. We fine-tune a pre-trained LLM $\pi_\theta$ via self-generated Chain-of-Thought (CoT) rationales on environment rewards.

that leaves a large part of the action space unexplored (up to 55%). Moreover, we observe that small-scale LLMs (2B) tend to copy the most frequent actions in the context regardless of their respective reward, which we refer to as frequency bias. In contrast, larger LLMs (27B) mostly diminish the frequency bias, yet they remain prone to greedy behavior at the cost of exploration. Similarly, we quantify the knowing-doing gap and find that LLMs often know how to solve a task (87% correct rationales) but fail at acting on this knowledge as they prioritize greedy actions (64% of actions when the rationale is correct).

To overcome these shortcomings, we propose Reinforcement Learning Fine-Tuning (RLFT) on self-generated CoT rationales. RL is the predominant learning paradigm in decision-making scenarios and has been successful in game-playing (Silver et al., 2016; Vinyals et al., 2019), robotics (Tirumala et al., 2025), plasma-control (Degrave et al., 2022), or navigating stratospheric balloons (Bellemare et al., 2020). We study the effects of RLFT on pre-trained Gemma2 models (Team et al., 2024b;c) in three sizes (2B, 9B, 27B) in multi-arm bandit (MAB) and contextual bandit (CB) settings proposed by Nie et al. (2024), and the textual Tic-tac-toe environment released by Ruoss et al. (2024). Across environments, we find that RLFT enhances the decision-making abilities of LLMs by increasing exploration and narrowing the knowing-doing gap. While RLFT positively affects exploration of LLM agents, their exploration strategies remain sub-optimal. Therefore, we empirically evaluate both "classic" exploration mechanisms commonly employed in RL, such as $\epsilon$-greedy, and LLM-specific approaches, such as self-consistency, to enable more effective fine-tuning for decision-making. Finally, in our ablations, we investigate the importance of CoT reasoning for decision-making, highlight the effectiveness of leveraging expert data, and show the benefits of giving the agent more reasoning tokens to solve the decision-making problem. We emphasize that the central aim of this work is to provide an in-depth analysis of LLM behavior in agentic scenarios, illustrating potential shortcomings and avenues for improvement. We do not argue or believe that solely improving performance via RLFT is a complete solution to the shortcomings outlined.

In summary, we make the following **contributions**:

- We systematically examine three failure modes of small-to-medium scale LLMs in decision-making scenarios: greediness, frequency bias, and the knowing-doing gap.
- We study how RL fine-tuning on self-generated CoT rationales affects these shortcomings, highlighting positive effects of RLFT on exploration and decision-making abilities.
- We evaluate a variety of exploration mechanisms (e.g., $\epsilon$-greedy) and LLM-specific approaches (e.g., self-consistency), to enable more effective RLFT for LLMs.

## 2   RELATED WORK

**Exploration in RL and LLMs.** The trade-off between *exploration* and *exploitation* is a long-standing challenge in the field of RL (Schmidhuber, 1991a;b; Still & Precup, 2012; Oudeyer et al., 2007). Widely used RL agents have often relied on random schemes (Mnih et al., 2015), heuristics such

as state-visitation counts (Ecoffet et al., 2019; Raileanu & Rocktäschel, 2020), intrinsic curiosity (Pathak et al., 2017; Burda et al., 2018; Groth et al., 2021), behavior priors (Rao et al., 2021), or maximum entropy regularization (Schulman et al., 2017; Haarnoja et al., 2018). Naturally, a number of works looked into leveraging LLMs for improving exploration of RL agents either as a source of rewards (Klissarov et al., 2023; Lu et al., 2024) or to orchestrate exploration strategies (Klissarov et al., 2024). Krishnamurthy et al. (2024) investigate the in-context exploration abilities of LLMs when acting directly as a policy. Similarly, Nie et al. (2024) study the exploration abilities of LLMs when fine-tuned on expert trajectories. In contrast, our work particularly investigates the effects of RLFT on the exploration abilities of LLMs and focuses on *why* models fail.

**In-context Learning for Decision-Making.** ICL is a form of Meta-learning, also referred to as learning-to-learn (Schmidhuber, 1987). While meta-learning is targeted via a meta-training phase (Santoro et al., 2016; Mishra et al., 2018; Finn et al., 2017; Wang et al., 2016; Duan et al., 2016; Kirsch et al., 2019; Flennerhag et al., 2019; Team et al., 2023a), ICL emerges as a result of the pre-training data distribution (Chan et al., 2022; Kirsch et al., 2022). ICL has been rediscovered in LLMs (Brown et al., 2020a) after initial observations by Hochreiter et al. (2001) in LSTMs (Hochreiter & Schmidhuber, 1997). Mirchandani et al. (2023) leverage the ICL abilities of LLMs to operate as general pattern machines. A number of works leverage the CoT abilities (Wei et al., 2022) of LLMs in simple text-based scenarios (Shinn et al., 2023; Yao et al., 2022). Similar in-context decision-making abilities have been observed in models trained from scratch, albeit in restricted environments (Laskin et al., 2022; Lee et al., 2022; Kirsch et al., 2023; Raparthy et al., 2023; Schmied et al., 2024b;a).

**Self-Correction in LLMs.** A critical component for LLM agents is the ability to self-correct over previously explored attempts. Existing works focus primarily on math benchmarks (Cobbe et al., 2021; Hendrycks et al., 2021; Welleck et al., 2022). Zelikman et al. (2022) leverage hints to iteratively generate correct answers and fine-tune on the respective CoT rationales. Kumar et al. (2024) employ RLFT over multiple trials to induce self-correction. Similarly, Zelikman et al. (2024) make use of RL fine-tuning, but instead generate rationales at every token position. Instead of imitation, Wang et al. (2025) rely on critique fine-tuning to induce self-correction. Wulfmeier et al. (2024) make use of inverse RL to avoid compounding errors. Other works rely on ICL abilities to learn from previous mistakes (Zhang et al., 2024; Monea et al., 2024). In contrast to Monea et al. (2024), who use ICL and contexts that span multiple previous episodes, we rely on single-trajectory contexts and CoT reasoning abilities. Moreover, we focus on analyses of the decision-making abilities of LLMs and the effects of RLFT, rather than proposing a new method. While conceptual corrections are possible, exact token-level correction is usually difficult for autoregressive generation (Cundy & Ermon, 2023).

## 3 METHODOLOGY

### 3.1 BACKGROUND

**Reinforcement Learning.** We assume the standard RL formulation via a Markov Decision Process (MDP) represented by a tuple of $(\mathcal{S}, \mathcal{A}, \mathcal{P}, \mathcal{R})$, where $\mathcal{S}$ and $\mathcal{A}$ denote state and action spaces, respectively. At every timestep $t$ the agent observes state $s_t \in \mathcal{S}$, predicts action $a_t \in \mathcal{A}$, and receives a reward $r_t$ given by the reward function $\mathcal{R}(s_t, a_t)$. $\mathcal{P}(s_{t+1} \mid s_t, a_t)$ defines the transition dynamics constituting a probability distribution over next states $s_{t+1}$. The goal of RL is to learn a policy $\pi_\theta(a_t \mid s_t)$ with parameters $\theta$ that predicts an action $a_t$ in state $s_t$ that maximizes cumulative reward.

**Reinforcement Learning from Human Feedback.** RLHF aims to fine-tune pre-trained models towards human preferences (Christiano et al., 2017). Preferences are typically encoded via a reward model $r_\phi$ with parameters $\phi$ learned from a human-annotated dataset $\mathcal{D}$ consisting of query-response pairs $x$ and $y$, respectively. RLHF optimizes a constrained REINFORCE estimator (Williams, 1992):

$$\max_\theta \mathbb{E}_{x \sim \mathcal{D}, y \sim \pi_\theta(\cdot|x)} \left[ (r_\phi(x, y) - b)\nabla_\theta \log \pi_\theta(y \mid x) - \beta D_{KL}(\pi_\theta(\cdot \mid x) \mid\mid \pi_{ref}(\cdot \mid x)) \right] \quad (1)$$

Here $\pi_{ref}$ is a reference policy, which is typically the frozen pre-trained model, and $\beta$ is a weighting term. The baseline $b$ represents a baseline to reduce variance and is commonly instantiated by a value function (Schulman et al., 2017; Ouyang et al., 2022) or a Monte-Carlo (MC) estimate of the returns (Ahmadian et al., 2024; Ramesh et al., 2024; Shao et al., 2024).

## 3.2 Reinforcement Learning Fine-Tuning (RLFT)

Our RLFT approach relies on fine-tuning on self-generated CoT rationales on rewards obtained from environment interaction. During RLFT, the model learns to iteratively refine its reasoning process, favoring CoT patterns and actions that lead to higher rewards (see Figure 1). Our approach is akin to Guo et al. (2025) and targets decision-making scenarios similar to Zhai et al. (2025).

**Context Representation.** The input tokens to our model at step $t$ consists of input instructions $c_t^{in}$, output instructions $c_t^{out}$, and the most recent interaction history $c_t^{\tau_{t-C:t}}$ (see Figure 1). The history representation contains the trajectory $\tau_{t-C:t} = (s_{t-C}, a_{t-C}, r_{t-C}, \ldots, s_t, a_t, r_t)$ of the $C$ most recent states, actions, and rewards. We opt for task-specific instructions for $c_t^{in}$ rather than a generic instruction template, providing the agent with information about the observations, the possible actions, and its objective. Consequently, $c_t$ is represented by the concatenation of the instruction and history tokens $c_t = [c_t^{in}; c_t^{out}; c_t^{\tau_{t-C:t}}]$.

**Factorization of Action Tokens.** At every interaction step $t$, the agent generates action tokens $z_t = [z_t^{CoT}; a_t]$ containing both the CoT reasoning tokens $z_t^{CoT}$ and the action to be executed in the environment $a_t$. To extract $a_t$ from $z_t$, we make use of an extraction function $a_t = g(z_t)$. In practice, $g$ consists of regular expressions to match the output pattern given by $c_t^{out}$. If no valid action is found, a random action is executed. To allow for flexibility in refining the reasoning process, we opt for a permissive output template (i.e., ACTION=X), rather than enforcing a structured output template (e.g., <action> blocks). We employ a token generation budget of $G$ tokens (=256), therefore $|z_t| \leq G$.

**Reward Shaping for Valid Actions.** In addition to the environment reward $r_t^{env}$, we employ a reward shaping term $r_t^{valid}$ to encourage the model to adhere to the output template, $r_t = r_t^{env} + r_t^{valid}$. More specifically, we make use of a reward penalty of $-5$ if $g$ cannot extract a valid action, $r_t^{valid} = -5 \cdot \mathbb{1}(g(a_t^{act}) \notin \mathcal{A})$. To ensure that the reward penalty does not overly bias optimization, we employ reward normalization to the environment rewards.

**Fine-tuning objective.** We fine-tune using the clipping objective introduced by Schulman et al. (2017) with and additional KL constraint to the reference policy $\pi_{ref}$:

$$\max_\theta \mathbb{E}_{(c,z)\sim\mathcal{D}} \left[ \min\left( \frac{\pi_\theta(z|c)}{\pi_{\theta_{old}}(z|c)} A_{adv}, \text{clip}_\epsilon\left( \frac{\pi_\theta(z|c)}{\pi_{\theta_{old}}(z|c)} \right) A_{adv} \right) - \beta D_{KL}(\pi_\theta(\cdot|c)||\pi_{ref}(\cdot|c)) \right] \tag{2}$$

Here $\pi_{\theta_{old}}$ refers to the rollout generating policy, $D$ is the rollout buffer, and $\epsilon$ is a hyperparameter. To allow for memory-efficient fine-tuning in environments with fixed episode lengths (bandits), we make use of a Monte Carlo baseline to estimate $A_{adv}$. Instead of exploiting multiple rollouts, as used by Ahmadian et al. (2024) and Ramesh et al. (2024), we compute rewards-to-go. For environments with variable episode lengths (Tic-tac-toe), we learn a separate state-value head on top of the last layer LLM representations and make use of generalized advantage estimation (Schulman et al., 2015) (see Appendix B) We provide additional implementation and training details in Appendix B.

## 4 LLMs for Decision-Making

We study the effect of fine-tuning LLMs in MABs and CBs settings proposed by Nie et al. (2024), and on a text-based version of Tic-tac-toe released by Paglieri et al. (2024). For our experiments, we primarily focus on Gemma2 (Team et al., 2024c) at three model scales (2B, 9B, and 27B) and report analyses for Llama3 (Dubey et al., 2024) and Qwen2.5 (Qwen et al., 2025) in Appendix C.4. In Section 4.2, we first analyze three common failure modes of LLM agents in MABs: **(1)** greediness, **(2)** frequency bias, and **(3)** the knowing-doing gap. Then we investigate the effects of fine-tuning on self-generated CoT rationales or expert rationales in MABs/CBs (Section 4.3), and in Tic-tac-toe (Section 4.5). In Section 4.4, we study the effects of exploration mechanisms on the fine-tuning performance. Finally, in Section 4.5 we empirically examine important components of our approach.

### 4.1 Environments & Baselines

**Multi-armed and Contextual Bandits.** MABs (Slivkins et al., 2019; Lattimore & Szepesvári, 2020) are a classic problem setting in RL that isolates the *exploration-exploitation* trade-off. For our MAB experiments, we leverage the text-based bandit scenarios released by Nie et al. (2024). We focus

on the *continuous* and *button* variants, as illustrated in Figure 2. We report results for MAB with $k \in \{5, 10, 20\}$ arms ($|\mathcal{A}| = k$) and payoffs of the arms being either Gaussian or Bernoulli distributed. In addition, we consider three levels of stochasticity (low/medium/high) that determine the standard deviation or delta gap in Gaussian or Bernoulli bandits, respectively. For all MAB settings, we limit the horizon $T$ to 50 interaction steps. We provide more details on our MAB setup in Appendix A.1.

We compare against two commonly used baselines for MABs: Upper-confidence Bound (UCB) (Auer, 2002) and a random agent that selects actions uniformly at random. UCB is considered optimal and represents the upper bound for agent performance, whereas the random baseline represents the lower bound. We want to emphasize that we do not aim to outperform UCB with LLMs, but instead aim for a better understanding. We provide more details on our baselines and on our setup for CBs in Appendices A.1 and A.2, respectively.

**Tic-tac-toe.** In addition, we use the text-based Tic-tac-toe environment released by Ruoss et al. (2024), which exhibits proper state transitions. Ruoss et al. (2024) demonstrated that frontier models struggle to achieve strong performance in this environment and often barely beat a random opponent. Consequently, it is a good target to investigate the efficacy of RLFT. In Appendix A.3, we provide additional details on our Tic-tac-toe environment and training setup.

> **Button Multi-armed Bandit (Gaussian)**
>
> You are a bandit algorithm in a room with 5 buttons labeled red, green, blue, yellow, orange. [...].
>
> Think step-by-step and output your final answer in the format ACTION=X where X is one of the arms listed above. IMPORTANT: Provide your (SHORT!) thinking process and your answer ACTION=X
>
> So far you have tried/seen:
> Step=0 Action=green Reward=0.3
> Step=1 Action=blue Reward=0.1
> Step=2 Action=orange Reward=-0.5
> ...
> What do you predict next?

**Figure 2:** *Button* MAB from (Nie et al., 2024) using our context representation and instructions.

## 4.2 WHY DO LLMS PERFORM SUBOPTIMALLY IN DECISION-MAKING?

Prior works found that LLM agents often perform suboptimally and fail to explore sufficiently in interactive settings (Paglieri et al., 2024; Ruoss et al., 2024). Therefore, we first examine *why* models perform suboptimally and identify three prevalent failure modes: **(1)** greediness, **(2)** frequency bias, and **(3)** the knowing-doing gap. In this section, we present analyses of LLMs models when given input contexts that elucidate the failure corresponding modes. We conduct our analyses on the *button* instance of our MAB experiments at three model scales, and find that the failure modes persist across model scales (see Appendix C.1 for additional results on a *continuous* MAB instance).

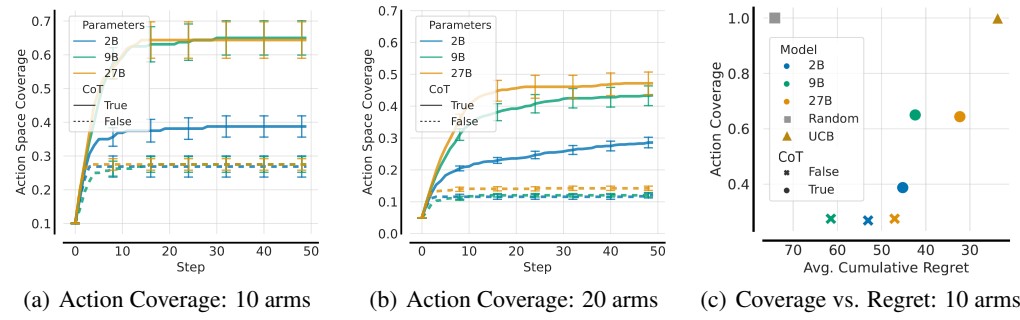

(a) Action Coverage: 10 arms      (b) Action Coverage: 20 arms      (c) Coverage vs. Regret: 10 arms

**Figure 3:** Illustration of **greediness**. We show action coverage for Gemma2 2B/9B/27B w/ and w/o CoT for **(a)** 10 and **(b)** 20 arms over 50 interaction steps. Agents favor the best-performing action among the set of selected actions, leading to stagnating action coverage, despite benefits of larger models and CoT. In **(c)**, we plot cumulative regret against action coverage. The agents exhibit suboptimal regret because of greedy action selection strategies.

**Greediness.** The first and most pervasive failure mode is *greediness*, which is characterized by the LLM overly favoring the best-performing action among a small set of actions seen so far. To illustrate this failure mode, we show the average action coverage achieved by Gemma2 2B/9B/27B with and without CoT across 64 MABs with 10 and 20 arms over 50 interaction steps (see Figure 3 a and b). We define **action coverage** $C_t$ at step $t$ as the fraction of available actions that have been selected at least once, $C_t = \frac{\{a \in \mathcal{A}: N_t(a) > 0\}}{|\mathcal{A}|}$ with $N_t(a)$ representing the number of times action $a \in \mathcal{A}$ has been selected until $t$. For 10 arms and averaged over 64 parallel environments, we find that Gemma2 2B covers $40\%$ of all actions, while 9B/27B cover $65\%$ (i.e., 6.5 actions), leaving a significant part of the action space unexplored. Note that without CoT, all models explore merely $25\%$ of all actions in the 10 arms setting. The suboptimal coverage is caused by the model overly favoring high-reward actions (see Figure 15 in Appendix C.1.1). Consequently, the model prematurely commits to a greedy strategy, leading to a stagnating action coverage beyond 10 steps. Increasing the number of arms makes the greediness even more apparent, with the largest models only covering $45\%$ of all actions. Due to this, the regret remains high compared to UCB, even though the models improve significantly over a random agent (see Figure 3c). We repeat our analyses with the Llama3 (Dubey et al., 2024) and Qwen2.5 (Qwen et al., 2025) model families in Appendix C.4 and show that these biases persist.

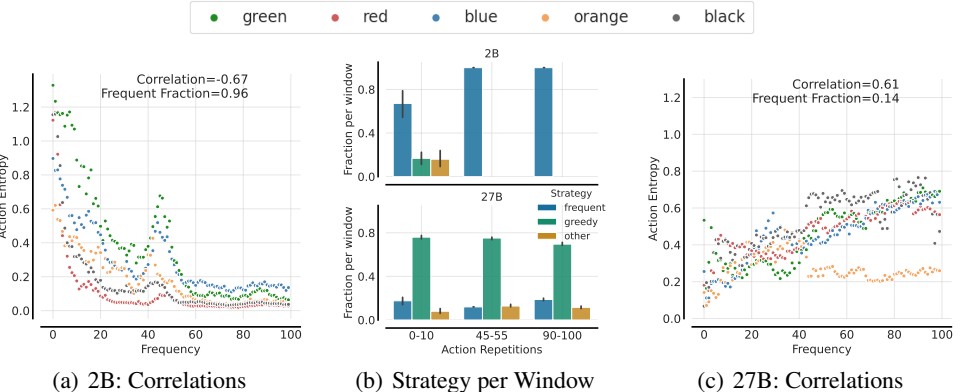

|  |  |  |
|---|---|---|
| (a) 2B: Correlations | (b) Strategy per Window | (c) 27B: Correlations |

**Figure 4:** Illustration of **frequency bias**. We plot the frequency of the repeated action in the context against the action entropy across all actions for 10 armed MABs in the button scenario (actions are colors). **(a)** Gemma2 2B heavily suffers from frequency bias, becoming more certain of the most frequent action, the more often it occurs in the context. **(c)** Gemma2 27B overcomes the frequency bias, but instead behaves greedily. In **(b)**, we show the action strategies for three repetition windows.

**Frequency Bias.** The next prevalent failure mode is *frequency bias*, characterized by repeatedly selecting the most frequent action in the context, even when that action gives low reward. To understand how the model's behavior is influenced by the frequency, we construct prefix histories using a random policy, vary the number of repetitions of the last action in the history (0-100), and record the entropy over all actions (Figure 4a and c). See Appendix C.1.2 for details on the context generation. To quantify frequency bias, we categorize an action as **frequent** $a_f = \arg\max_{a \in A} N_T(a)$, **greedy** $a_g = \arg\max_{a \in \{a \in A: N_T(a) > 0\}} R_T(a)$, or **other** if they are neither frequent nor greedy. Note that an action is optimal with 10% probability. Subsequently, we compute the frequent $F_f$, greedy $F_g$, and other $F_o$ fractions as reported in Figure 4 (see Appendix 4 for additional definitions).

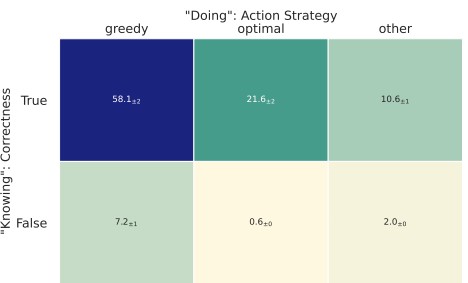

**Figure 5:** Confusion matrix for the **knowing-doing gap** of Gemma2 27B. The agent "knows" how to solve the task (87% correct rationales, sum of top row), but fails at "doing" (58% greedy actions among correct rationales). See Figure 26 for the CoT instructions and an agent response.

Gemma2 2B heavily suffers from repeated actions, exhibiting a decreasing entropy with increasing repetitions (96% $F_f$, see Figure 4a). In contrast, 27B

escapes the frequency bias (14%, see Figure 4c) and interestingly becomes less certain of its action prediction with increasing repetitions. To examine this further, we show the bucketized fractions with 0-10, 45-55, and 90-100 repetitions for 2B and 27B in Figure 4b. Indeed, for 2B $F_f$ keeps increasing with increasing repetitions. While 27B escapes the frequency bias, it suffers heavily from greediness. Similar biases have been identified in Behavior Cloning (BC) settings using small models and termed *copycat* bias (Wen et al., 2020; Schmied et al., 2024b). This suggests that the persistent frequency bias in smaller models may be an artifact of supervised pre-training or insufficient model capabilities, and motivates the use of RL as a counter-measurement.

**Knowing-Doing Gap.** The *knowing-doing gap* has been observed by Paglieri et al. (2024) and Ruoss et al. (2024). Our simple MAB setting is well-suited to quantify the knowing-doing gap precisely. To investigate this gap in our setting, we first task Gemma2 27B to produce the UCB algorithm, to compute the relevant quantities accordingly ("knowing"), and finally to act according to the computed quantities ("doing", see Figure 26 for the instructions and an agent response). We let Gemma2 27B interact with the environment (64 instances) for 50 timesteps with $G = 2048$ per step, and extract the UCB quantities from the generated rationales. To quantify ***"knowing"***, we compare the UCB values computed by the model against the real UCB values, and consider the rationale $z_{CoT}$ as correct if the arm with the highest UCB values matches (see Appendix C.1.3 for details). To quantify ***"doing"***, we categorize the generated actions as *optimal* action if the model selects the action with the highest UCB value, as *greedy* if it selects the action with the highest UCB value among the set of actions tried so far, and as *other* if the action is neither optimal nor greedy. Subsequently, we compute the percentages of greedy/optimal/other actions. The agent clearly knows how to solve the task, with 87% of all rationales being correct (see Figure 5). However, even for correctly computed rationales, the model often selects the greedy action (58%) over the optimal action (21%). This discrepancy highlights the shortcomings of the LLM when it comes to "acting" even when "knowing" the algorithm.

## 4.3 EFFECTIVENESS OF RL FINE-TUNING

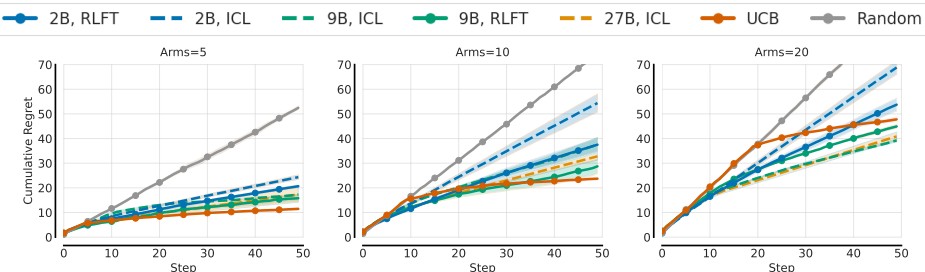

**Figure 6:** Main Comparison on **Gaussian MABs** button scenario in the medium noise ($\sigma = 1$) setting. We compare cumulative regrets (lower is better) of classic baselines against ICL and RLFT performances for 5, 10, and 20 arms. See Figure 20 for $\sigma = 0.1$ and $\sigma = 3$.

Next, we study the effects of RLFT on cumulative regret (w.r.t. optimal policy) and whether it alleviates the highlighted failure modes. We fine-tune Gemma2 2B and 9B on self-generated CoT rationales for 30K updates with an (accumulated) batch size of 128. To avoid memorization of reward distributions, we maintain a pool of 512 MABs and randomly select a subset of 16 MABs per rollout during training, and report evaluation results on hold-out MABs (see Appendix B for details).

**RLFT lowers regret.** In Figure 6, we report the cumulative regrets across model sizes and arms for a medium noise $\sigma = 1.0$ scenario (see Appendix C.2 for low/high noise). Across environments, the LLMs clearly outperform the random baseline, and RLFT lowers regret for both 2B and 9B. For 2B, RLFT narrows the gap to its larger counterparts and UCB. Similarly, RLFT lowers regret for Gemma2 9B. Note that the lower cumulative regret of Gemma2 9/27B compared to UCB after 50 environment steps in the 20 arms scenario is an artifact of the limited interaction steps, but the trends remain clear. We repeat RLFT for CBs, and observe similar performance improvements for Gemma2 2B (see Appendix C.3). Consequently, reinforcing self-generated CoT rationales towards environment rewards improves performance on simple decision-making scenarios.

**RLFT mitigates greediness.** In Figure 7, we report the action coverage for 2B after RLFT at different numbers of gradient steps (10K, 20K, 30K). Indeed, we observe that RLFT results in increased action coverage (+12%) after 30K updates. Interestingly, we first observe a decrease (at 10K) followed by an increase in action coverage (20K, 30K). We observe similar effects for the 20 arms scenario (see Figure 17). Via RLFT, the agent learns to explore its options and to mitigate greediness.

**RLFT counteracts frequency bias.** We find that RLFT counteracts frequency bias (see Figure 19). In particular, for 0-10 repetitions, we observe a strong decrease in the fraction of frequent actions (70% → 35%) and an increase in "other" actions (8% → 35%). However, $F_f$ remains elevated for high repetitions. Consequently, RLFT counteracts frequency bias, but does not fully alleviate it.

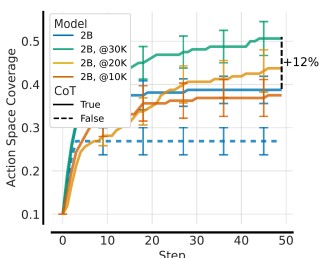

**Figure 7:** Effect of RLFT on greediness for Gemma2 2B.

### 4.4 Effects of Exploration Mechanisms

For RLFT, we relied solely on the exploration properties for CoT reasoning. However, in RL it is common practice to employ additional exploration strategies (Mnih et al., 2015; Schulman et al., 2017; Haarnoja et al., 2018). Therefore, we study the effects of classic exploration mechanisms and LLM-specific strategies to encourage exploration. We compare: (1) try-all actions initially similar to UCB, (2) $\epsilon$-greedy, (3) context randomization, (4) context summary similar to Krishnamurthy et al. (2024) and Nie et al. (2024), (5) self-correction similar to Kumar et al. (2024), (6) self-consistency (Wang et al., 2022), and (7) incorporating an exploration bonus during RLFT. We provide details on each mechanism in Appendix B.4. Across model scales, we observe that the mechanisms result in varied effects on action coverage (see Figure 8). First, we find that the simple *try-all* strategy, which reduces the need for additional exploration by trying all actions, results in large performance improvements. This suggests that only given sufficient information about the (sub-)optimality of actions, LLMs are able to select actions accordingly, underscoring their exploration shortcomings. Second, a simple *exploration bonus* (+1 reward for untried actions) during RLFT significantly increases exploration (50% → 70%) and lowers regret towards the expert compared to regular RLFT. This highlights the importance of reward shaping for fine-tuning LLMs to elucidate a desired behavior.

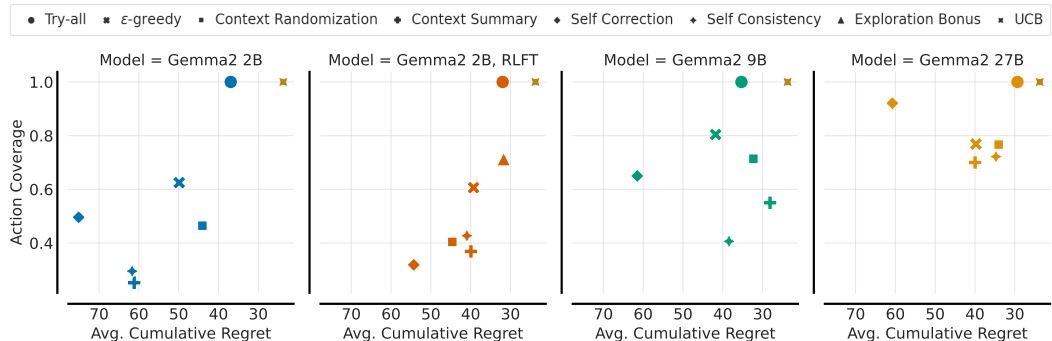

**Figure 8:** Effect of **exploration mechanisms** on action coverage and cumulative regret.

### 4.5 Ablations

**RLFT in Tic-tac-toe.** To investigate the efficacy of RLFT in stateful environments, we evaluate on Tic-tac-toe from Ruoss et al. (2024), in which frontier models struggle to achieve strong performance (see Appendix B for training details). We fine-tune against three opponents: a random agent, Monte Carlo Tree Search (MCTS) (Coulom, 2006), and noisy MCTS (50% of actions selected at random). We find that RLFT significantly enhances the win-rate of Gemma2 2B against all opponents compared to ICL (see Figure 9a). Against the random agent, RLFT elevates the average return from 0.15 (i.e., winning 15% of games) to 0.75. Notably, the agent even manages to draw against the optimal MCTS

baseline ($-0.95 \rightarrow 0.0$), underscoring the effectiveness of RLFT for decision-making. However, for high performance, it is essential to provide the currently legal actions in the context (see Figure 25).

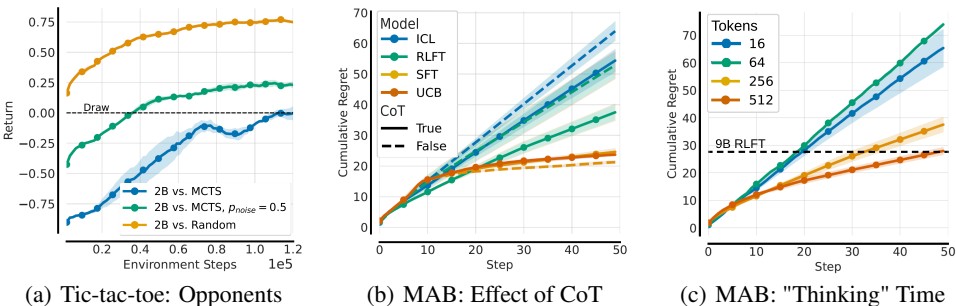

|  (a) Tic-tac-toe: Opponents | (b) MAB: Effect of CoT | (c) MAB: "Thinking" Time |

**Figure 9:** Ablations. **(a)** Effect of RLFT in Tic-tac-toe from Ruoss et al. (2024). **(b)** Effect of CoT on ICL, RLFT and SFT (expert data) performance on MABs. **(c)** Effect of increasing the number of "thinking" tokens to generate during RLFT.

**Importance of CoT for RLFT.** CoT reasoning is critical for ICL performance (see Figure 3), but the question remains how CoT influences RLFT. Therefore, we run RLFT on Gemma2 2B on the 10 arms Gaussian MAB both with and without CoT (see Figure 9b, RLFT). Indeed, without CoT, RLFT barely attains the performance of ICL with CoT. This highlights the function of CoT as a vital exploration and rationalization mechanism for decision-making.

**Expert Behavior Cloning vs. Thought Cloning.** BC is a prevalent approach in sequence models for decision-making (Pomerleau, 1988; Brohan et al., 2022; 2023) and relies on expert datasets. Consequently, we construct two UCB expert datasets comprising 32K rollouts (1.6M transitions) across different MABs either with or without CoT traces (see Figure 12 for an illustration) and perform SFT on them. Notably, both SFT variants successfully mimic the expert, achieving comparable regret to UCB (see Figure 9b, SFT). This result underscores the effectiveness of expert data in decision-making, echoing findings in reasoning tasks (Muennighoff et al., 2025).

**Effect of Thinking Time.** Finally, we investigate the effect of giving the agent more/less time to "think" during RLFT by varying the generation budget $G$ (see Figure 9c and Appendix D). Decreasing $G$ results in poor performance, as the agent is unable to rationalize its decisions. Increasing $G$ to 512 improves performance to the level of 9B w/ RLFT. The agent effectively leverages the additional tokens, which reflects recent observations in mathematical reasoning (Guo et al., 2025). However, when increasing $G$, rollout generation can make up the majority of the training time due to the multi-step nature of decision-making tasks (e.g., for $H = 50$, $G = 500$, agent generates 25K tokens).

## 5 CONCLUSION

We study *why* LLMs perform sub-optimally in decision-making scenarios and examine three prevalent and independent failure modes: greediness, frequency bias, and the knowing-doing gap. We emphasize that we focus on in-depth analyses to provide a fundamental understanding of LLM behavior in decision-making scenarios and to aid future research, rather than solely improving algorithm performance via RLFT. In particular, our rigorous empirical analysis aims to inform design decisions of future LLM agents. For example, when building tool-use or coding agents, it may be essential to restrict the breadth of available tools initially due to pre-existing greediness biases (Yang et al., 2025). We show that RLFT on CoT rationales mitigates greediness, counteracts frequency bias, and improves performance. While RLFT improves exploration abilities, it remains sub-optimal compared to bandit algorithms. Therefore, we investigate a variety of mechanisms to enhance exploration. Models act near-optimally if provided with sufficient information, underscoring their shortcomings in exploration. Finally, we highlight the importance of reward shaping for RLFT.

**Future Work.** We primarily focused our evaluation on the Gemma2/Llama3/Qwen2.5 series and small-to-medium scale models. While we expect that our findings transfer to larger models, we deem research into frontier models important. Moreover, our MAB experiments were conducted with a

limited horizon of 50 environment steps, which is sufficient for 5 and 10 arms, but insufficient for 20 arms. Due to its simplicity, the MAB setting also allows us to rigorously study how the learned decision-making and exploration abilities transfer to unseen environments in future work (e.g., by varying the number of arms or shifting the reward distribution at test time). We expect that RLFT may increase robustness compared to SFT in such scenarios, which would reflect findings by Chu et al. (2025). For future work, we believe that evaluating the exploration abilities of LLM agents is particularly interesting in environments that require targeted exploration towards an end-goal. First, this includes other stateful environments from Paglieri et al. (2024), Ruoss et al. (2024), and Tajwar et al. (2025). A deeper investigation into the knowing-doing gap would be particularly interesting in these environments. Second, we deem a systematic investigation into the exploration abilities of LLMs in existing agentic and computer-use benchmarks (Mialon et al., 2023; He et al., 2024; Zhou et al., 2023) interesting. While RLFT positively influenced the agent's decision-making abilities, our experiments demonstrated that performance remained suboptimal. Simple exploration mechanisms, such as an extrinsic reward bonus, considerably improved performance. Similarly, intrinsic reward mechanisms (Pathak et al., 2017; Burda et al., 2018) to encourage autonomous exploration could be valuable components of future agents in multi-turn applications. Moreover, in our ablations, we found that LLMs benefit from additional "thinking" time and believe that allowing for a larger generation budget is becoming increasingly important for agentic scenarios, especially for scenarios that involve high-stakes decisions (e.g., economics or AI safety). We deem investigations into such high-stakes scenarios fruitful for future work. While increasing "thinking" time improves performance, it comes with excessive computational cost at training time due to the rollout generation and the multi-step nature of decision-making. Therefore, modern recurrent architectures (Gu & Dao, 2023; De et al., 2024; Beck et al., 2025) that allow for faster inference may be promising alternatives for decision-making and scaling-up RLFT.

## REPRODUCIBILITY STATEMENT

To ensure reproducibility, we provide comprehensive details in the Appendix. We describe our environment setup, the generated datasets, and our baselines in Appendix A. The primary benchmark we focus on, BanditBench, was released by Nie et al. (2024) and is available on GitHub. Furthermore, in Appendix B, we provide the implementation details for all RLFT and SFT experiments, along with a comprehensive list of hyperparameters. Moreover, in Appendix B.4, we describe and provide details for the exploration mechanisms compared in Section 4.4. Finally, in Appendix C, we provide additional experimental results that we could not fit in the main text.

## USE OF LARGE LANGUAGE MODELS

In preparing this manuscript, we used LLMs to polish existing text, specifically for improving grammar and some phrasings. No LLMs were used for research ideation or exploration.

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

CONTENTS

## A  ENVIRONMENTS & DATASETS

We conduct experiments on three sets of environments: multi-armed bandits, contextual bandits, and Tic-tac-toe. For the SFT experiments reported in Section 4.5, we generate our own expert datasets. In this section, we provide additional details on our environments and datasets.

### A.1  MULTI-ARM BANDITS: BANDITBENCH

MABs (Slivkins et al., 2019; Lattimore & Szepesvári, 2020) are a classic problem setting in RL that isolates the *exploration-exploitation* trade-off. In contrast, commonly used RL environments (Bellemare et al., 2013; Tassa et al., 2018) often conflate exploration with other RL-specific aspects, such as delayed rewards (Arjona-Medina et al., 2019). We rely on the MAB scenarios released in BanditBench (Nie et al., 2024) and also used by (Krishnamurthy et al., 2024). MABs come with a number of variable dimensions, including the scenario type (textual description of the task), the type of reward distribution (Gaussian, Bernoulli) and its corresponding noise level (low/medium/high),

the number of arms (i.e., actions), and the number of interaction steps per episode. Consequently, MABs are a good testbed for LLM agents.

We focus on the *continuous* and *button* variants released by Nie et al. (2024) using their benchmark available on Github. We report results for MAB with $k \in \{5, 10, 20\}$ arms ($|\mathcal{A}| = k$) for three levels of stochasticity (low/medium/high). In our experiments, for every arm the corresponding reward is sampled from a Gaussian distribution $r \sim \mathcal{N}(\mu, \sigma)$ where $\mu \sim \mathcal{N}(0, 1)$ and is a fixed scalar $\sigma \in \{0.1, 1, 3\}$ for the three levels of stochasticity, respectively. For all MAB settings, we limit the horizon $T$ to 50 interaction steps. Limiting the horizon is necessary to handle the increasing lengths and, consequently, RAM requirements for fine-tuning. While we consider 50 interaction steps sufficient for 5 and 10 arms, it is insufficient for the 20-arms scenario. However, note that the general trends are well observable for the 20 arms scenario. In Figure 10, we show the continuous and button Gaussian MABs with CoT instructions for the agent. Similarly, in Figure 11 we show the same instances without CoT instructions.

---

**Continuous MAB (Gaussian)**

You are a bandit algorithm and interact with 5 arms labeled 0,1,2,3,4. Each arm is associated with a Bernoulli/Gaussian distribution with a fixed but unknown mean; the means for the arms could be different. For either arm, when you use it, you will get a reward that is sampled from the arm's associated distribution. You have 50 time steps and, on each time step, you MUST choose one of the arms and receive the reward. Your goal is to maximize the total reward.

[More Instructions]

Think step-by-step and output your final answer in the format ACTION=X where X is one of the arms listed above. IMPORTANT: Provide your (SHORT!) thinking process and your answer ACTION=X

So far you have tried/seen:
Step=0  Action=1  Reward=0.3
Step=1  Action=2  Reward=0.1
Step=2  Action=0  Reward=-0.5
Step=3  Action=3  Reward=0.5
Step=4  Action=1  Reward=0.24
...
What do you predict next?

---

**Button MAB (Gaussian)**

You are a bandit algorithm in a room with 5 buttons labeled red, green, blue, yellow, orange. Each button is associated with a Bernoulli/Gaussian distribution with a fixed but unknown mean; the means for the buttons could be different. For either button, when you press it, you will get a reward that is sampled from the button's associated distribution. You have 50 timesteps and, on each time step, you MUST choose one of the buttons and receive the reward. Your goal is to maximize the total reward over the 50 timesteps.

[More Instructions]

Think step-by-step and output your final answer in the format ACTION=X where X is one of the arms listed above. IMPORTANT: Provide your (SHORT!) thinking process and your answer ACTION=X

So far you have tried/seen:
Step=0  Action=green  Reward=0.3
Step=1  Action=blue  Reward=0.1
Step=2  Action=orange  Reward=-0.5
Step=3  Action=red  Reward=0.5
Step=4  Action=green  Reward=0.24
...
What do you predict next?

---

**Figure 10:** Illustration of *continuous* and *button* Gaussian multi-armed bandits scenarios from BanditBench (Nie et al., 2024) using our context representation and **with CoT** instructions.

### A.1.1 BASELINES

We compare against two commonly used baselines for MABs: Upper-confidence Bound (UCB) (Auer, 2002) and a random agent that selects actions uniformly at random. Generally, **UCB** aims to balance exploitation and exploration. The exploitation value in UCB:

$$V_t^{exploit}(a) = \sum_{t'=1}^{t} \frac{\mathbb{1}_{\{a_{t'}=a\}} r_{t'}}{N_t(a)} \tag{3}$$

for a given action $a$ at timestep $t$ represents the empirical mean of the rewards obtained when that action was executed, where $N_t(a)$ indicates the action frequency. The exploration bonus in UCB:

$$V_t^{explore}(a) = \alpha \sqrt{\frac{\log t}{N_t(a)}} \tag{4}$$

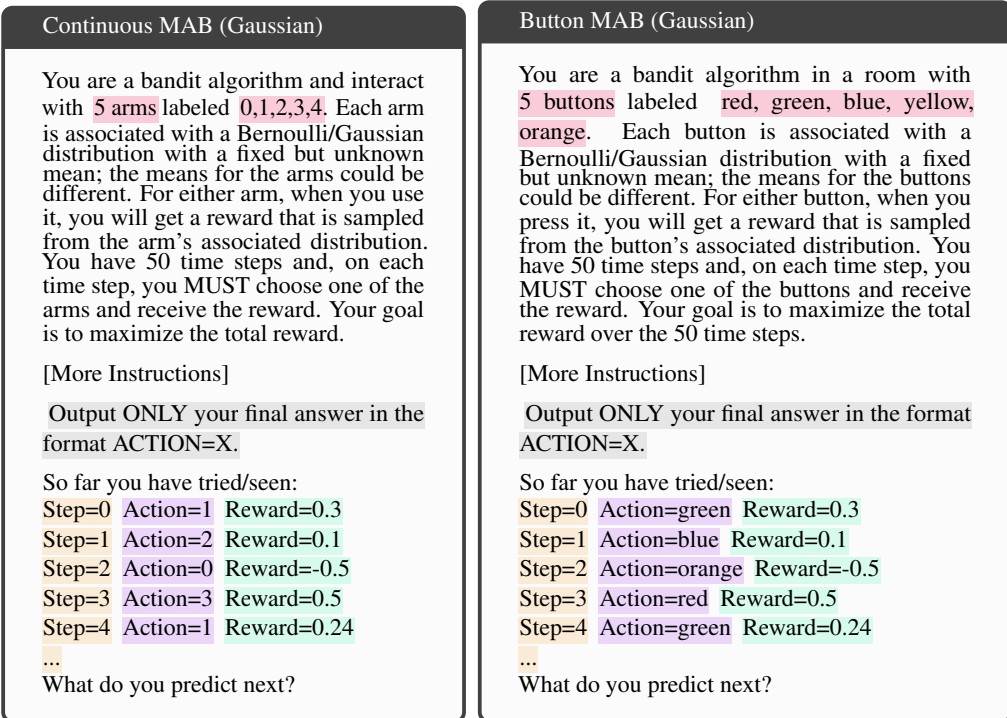

**Figure 11:** Illustration of *continuous* and *button* Gaussian multi-armed bandits scenarios from BanditBench (Nie et al., 2024) using our context representation **without CoT** instructions.

is proportional to how frequently that action $a$ has previously been selected, and $\alpha$ is a hyperparameter. UCB is considered optimal and represents the upper bound for agent performance, whereas the random baseline represents the lower bound. Consequently, we want to emphasize that the purpose of this work is not to outperform UCB or to propose a better bandit algorithm with LLMs and RLFT. Instead, we aim to better understand the decision-making abilities of LLMs and the effects of RLFT on exploration.

### A.1.2 SFT DATASETS

In our main experiments, we focused on self-generated CoT rationales and action predictions produced by our fine-tuned agents, which do not require a pre-collected dataset. In contrast, for our SFT experiments reported in Section 4.5, we generated UCB expert datasets. In particular, we construct two dataset instances: a *behavior cloning* dataset that only contains expert actions and a *thought cloning* (TC) dataset that incorporates expert actions alongside a thought process for coming up with the respective action (i.e., w/ CoT). For every arm and stochasticity level combination, we construct 32K rollouts from different MABs, which amounts to 1.6M transitions (state, action, reward) per dataset.

To provide the thought process in the TC datasets, we reconstruct the computations of the UCB values conducted by the UCB expert in textual format. The thought process ends with a final conclusion of why a particular action was selected (i.e., highest UCB value or exploratory action). Consequently, the action $z_t$ at step $t$ contains the thought process $z_{CoT}$ and the action to execute $a_t$ (see Section 3.2). We illustrate the actions contained in the dataset for a trajectory at steps 4 and 11 in Figure 12. The BC datasets do not contain the thought process. Instead, they only contain the final predictions made by the model, for example `ACTION=yellow` as shown in Figure 12.

### A.2 CONTEXTUAL BANDITS

MABs do not emit states. In contrast, CBs emit state representations at every interaction step, making them contextual. Consequently, CBs are interesting to test the abilities of LLMs to make use of the given context when predicting the next action. For our CB experiments, we leverage the MovieLens environment released by Nie et al. (2024), a semisynthetic bandit task based on the MovieLens

---

**Multi-armed Bandit at step 4**

Let's think step-by-step. We calculate the counts and means for every action.

[More Thoughts]

Count for action blue = 1, Mean = (-1.91) / 1 = -1.91, UCB = -1.91 + sqrt(1 / 1)) = -0.91
Count for action green = 1, Mean = (1.41) / 1 = 1.41, UCB = 1.41 + sqrt(1 / 1)) = 2.41
Count for action red = 1, Mean = (0.45) / 1 = 0.45, UCB = 0.45 + sqrt(1 / 1)) = 1.45
Count for action yellow = 0, Mean = NaN, UCB = NaN
Count for action orange = 0, Mean = NaN, UCB = NaN

We have not yet selected all actions. Therefore, we select the next one.

ACTION=yellow

---

**Multi-armed Bandit at step 11**

Let's think step-by-step. We calculate the counts and means for every action.

[More Thoughts]

Count for action blue = 1, Mean = (-1.91) / 1 = -1.91, UCB = -1.91 + sqrt(1 / 1)) = -0.91
Count for action green = 4, Mean = (1.41 + 0.17 + 0.67 + -0.1) / 4 = 0.54, UCB = 0.54 + sqrt(1 / 4)) = 1.04
Count for action red = 3, Mean = (0.45 + 0.78 + 2.16) / 3 = 1.13, UCB = 1.13 + sqrt(1 / 3)) = 1.71
Count for action yellow = 1, Mean = (-1.03) / 1 = -1.03, UCB = -1.03 + sqrt(1 / 1)) = -0.03
Count for action orange = 1, Mean = (-1.2) / 1 = -1.2, UCB = -1.2 + sqrt(1 / 1)) = -0.2

We select actions according to the highest UCB value. Therefore, action red is selected.

ACTION=red

---

**Figure 12:** Illustration of UCB rationales contained in our **SFT expert datasets** at two timesteps (4 and 11) in the same trajectory. Both examples show the Thought Cloning dataset instance containing both the produced CoT rationale and the predicted action. The Behavior Cloning instances contain only the final action prediction (in red).

dataset (Harper & Konstan, 2015). In this setting, the agent operates as a movie recommendation engine given a contextual description of a user (10K users in total) and a list of $K$ possible movies. The context representation provides a textual description of the user to recommend the movie to. This description includes the user's gender, age, profession, location, and a numeric description of the user's preferences for each of the possible movies. As for MABs, we report results for $K \in \{5, 10, 20\}$, and limit the horizon to 50 interaction steps. In Figure 13, we provide an example for a MovieLens CB with 5 actions with our context representation and CoT instructions.

**Baselines.** Similar to MABs, we compare against LinUCB (Chu et al., 2011) and an agent selecting actions uniformly at random. We provide implementation details on our baselines in Appendix B.

A.3 TIC-TAC-TOE

Finally, we use the text-based Tic-tac-toe environment released by Ruoss et al. (2024) (see Figure 14 for an example). Unlike MABs and CBs, Tic-tac-toe is a stateful environment with proper state transitions (i.e., action predicted at step $t$ affects the state observed at step $t + 1$). The agent receives scalar rewards of 1, 0, and -1 for winning, drawing, and losing against its opponent, respectively. Episodes last until either of the players wins, draws, or loses. To enable easy extraction of actions from the generated rationales, we represent the action space as a discrete set of 9 actions, corresponding to the grid positions on the $3 \times 3$ grid used in Tic-tac-toe ($| \mathcal{A} |= 9$). However, only at the start of an episode are all 9 actions valid. Subsequently, only a subset is valid because of the currently taken board positions. We (optionally) provide the set of valid actions at a particular step in textual form in the context given to the agent. Ruoss et al. (2024) demonstrated that frontier models struggle to achieve strong performance in this environment and barely beat a random opponent. Consequently, we deem it a good target to investigate the efficacy of RLFT.

---

**MovieLens Contextual Bandit**

You are an AI movie recommendation assistant for a streaming platform powered by a bandit algorithm that offers a wide variety of films from different studios and genres. There are 5 unique movies you can recommend, named star_wars_(1977), contact_(1997), fargo_(1996), return_of_the_jedi_(1983), liar_liar_(1997). When a user visits the streaming platform, you assess their demographic description to choose a movie to suggest. You aim to match the user with movies they are most likely to watch and enjoy.

[More Instructions]

Think step-by-step and output your final answer in the format ACTION=X where X is one of the arms listed above. IMPORTANT: Provide your (SHORT!) thinking process and your answer AC-TION=X

So far you have tried/seen:

...

Step=4 This person is a 28-year-old man, working as an administrator and living in Santa Clara county, CA. The user has some numerical values that represent their true implicit preference or taste for all movies: [-0.04, 0.02, -0.02, -0.0, 0.02]
What do you predict next?

---

**Figure 13:** Illustration of **contextual MovieLens scenario** from BanditBench (Nie et al., 2024) using our context representation and instructions.

**Baselines.** Following Ruoss et al. (2024), we compare against a random agent by default. In addition, we also compare against (MCTS) (Coulom, 2006), and a noisy variant of MCTS that selects an action randomly with 50% chance and according to MCTS otherwise.

---

**Tic-tac-toe**

You are an agent playing tic-tac-toe. You observe a board with 9 entries that looks like this:
000
100
002
1 indicates that player 1 has placed a stone in that square. 2 indicates that player 2 has placed a stone in that square. 0 indicates that no stone has been placed in that square. You play as 1.
There are 9 possible actions: 0, 1, 2, 3, 4, 5, 6, 7, 8. The actions correspond to the following board locations
012
345
678

[More Instructions]

Think step-by-step and output your final answer in the format ACTION=X where X is one of the arms listed above. IMPORTANT: Provide your (SHORT!) thinking process and your answer ACTION=X

So far you have tried/seen:
Step=0 State=000000000 Action=0 Reward=0
Step=1 State=102000000 Action=4 Reward=0
Step=2 State=102010002 Action=5 Reward=0
What do you predict next?

---

**Figure 14:** Illustration of the text-based **Tic-tac-toe** environment.

## B    EXPERIMENTAL & IMPLEMENTATION DETAILS

### B.1    TRAINING & EVALUATION

In our experiments, we fine-tune Gemma2 models in three model sizes (2B/9B/27B). For all experiments, we use the instruction-tuned versions of Gemma2 and leverage the respective instruction pre- and post-fixes. For bandits, we fine-tune all models for a total of 30K updates and evaluate after every 10K steps. with an accumulated batch size of 128. Similarly, we fine-tune for 12K updates and evaluate every 4K updates on Tic-tac-toe. We report the mean and 95% confidence intervals over three seeds, as suggested by Agarwal et al. (2021).

**General.** We train all agents with an accumulated batch size of 128. We use a learning rate of $1e^{-4}$, 100 linear warm-up steps followed by a cosine decay to $1e^{-6}$. To enable memory-efficient fine-tuning of 2B and 9B models, we utilize the AdaFactor optimizer (Shazeer & Stern, 2018). We experiment with LoRA (Hu et al., 2022) for fine-tuning the 9B and 27B models, but found it insufficient for improving the agent's decision-making abilities in our setting. However, LoRA considerably reduces the amount of memory required for RLFT and has been shown to work well for supervised fine-tuning of decision-making agents (Schmied et al., 2023). Therefore, we deem it a promising candidate for RLFT in decision-making scenarios. Furthermore, we employ gradient clipping of 1.0. We list all hyperparameters in Table 1.

**Table 1:** Default **hyperparameters** used in our experiments.

| Name | Value | Description |
|---|---|---|
| **Training** | | |
| training_steps | 30K or 12K | Number of training steps. |
| eval_freq | 10K or 4K | Evaluation frequency (in updates). |
| batch_size | 128 | Accumulated batch size. |
| lr_scheduler | Linear + cosine | Learning rate scheduler |
| warmup steps | 100 | Warmup steps. |
| lr | $1e^4$ to $1e^6$ | Maximum learning rate. |
| optimizer | AdaFactor | Optimizer. |
| **Sequence Length & Generation Budget** | | |
| context_length | 1792 | Input context length. |
| num_tokens | 256 | Generation budget. |
| **RLFT** | | |
| rollout_steps | 800 or 2048 | Rollout steps in-between updates. |
| update_epochs | 1 or 2 | Update epochs over rollout-buffer. |
| reward_penalty | -5 | Reward penalty for invalid actions. |
| loss | PPO clipping objective + KL constraint | Objective function. |
| baseline | MC-baseline or state-value head | Baseline. |
| envs | 16 | Number of parallel envs. |
| $\epsilon$ | 0.2 | Clipping value. |
| $\beta$ | 0.05 | KL coefficient. |
| reward_norm | True | Whether reward normalization is used. |
| train_temp | 1.0 | Sampling temp during rollouts. |
| eval_temp | 0.0 | Sampling temp during evaluation. |
| top_p | 1.0 | Sampling top-p. |
| **Hardware** | | |
| accelerator | $8 \times$ H100 | Hardware accelerator. |

**Context Lengths & Generation Budget.** For all model sizes and tasks, we use a context length of 1792 for the input context. By default, we set the generation budget to 256 tokens, except for the knowing-doing gap analyses reported in Section 4.2, which require a larger budget of 2048 tokens, and our generation budget ablation. Consequently, the effective sequence length for fine-tuning is 2048.

**Hardware Setup.** We train all models on a server equipped with $8 \times$ H100 GPUs.

### B.2    RLFT

For our RLFT experiments on bandits, we employ the context representation, action factorization, reward shaping terms, and training objectives described in Section 3.2. To extract the target action $a_t$

from $z_t$, we make use of a stack of regex expressions against the target pattern (i.e., `ACTION=X`) and consider the last match in the generated tokens as $a_t$. In addition to being fairly robust, we found that this approach allows for more flexibility during the RLFT process and led to better outcomes than a more structured approach. We opt for this choice, as preliminary experiments showed it was more robust to varied, sometimes malformed CoT outputs, which was particularly important during early stages of RLFT. This design choice prioritized more reliable signal extraction during training over strict format adherence. However, we acknowledge that depending on the model at hand and its associated post-training approach, it makes sense to consider different established extraction approaches, such as `boxed{}`.

Across model sizes, we additionally found it essential to introduce a reward shaping term to penalize rationales that contain no valid actions. By default, we use a reward penalty of -5 for invalid actions. To ensure that the reward penalty does not overly bias the optimization, we employ reward normalization to the environment rewards. In preliminary experiments, we found that a range of values between $[-10, -1]$ resulted in stable training. In contrast, a high penalty of $-100$ caused instabilities. In case no valid action is found within the generated tokens, we execute a random action. While this occurs seldomly, this mechanism is a useful safeguard to ensure robust interaction with the environment during RLFT (see Table 2 for percentage of random actions). Empirically, we found that the reward shaping term is sufficient for the models to produce exclusively valid actions early on in the training. Consequently, this simple mechanism enables a consistent signal throughout RLFT, and we add it to our recipe.

We fine-tune using the clipping objective introduced by Schulman et al. (2017) with an additional KL constraint to the reference policy $\pi_{ref}$. We set $\beta = 0.05$ and $\epsilon = 0.2$ for all experiments. We make use of the approximated per-token KL divergence instead of computing the full KL. While we found that computing the full KL slightly improves performance, it slows down training considerably. In contrast to Ahmadian et al. (2024) and Ramesh et al. (2024), we do not rely on producing multiple rollouts, because it is impractical for the multi-step nature of decision-making tasks. We opt for this choice because obtaining multiple rollouts from the same state, as are required for GRPO (Ramesh et al., 2024) and RLOO (Ahmadian et al., 2024), is difficult in decision-making scenarios for two reasons. First, due to the multi-step nature of decision-making tasks, it would be necessary to execute $k$ different actions corresponding to $k$ rollouts at every interaction step. This is difficult due to the already high associated costs of generating tokens on every step. Incorporating multiple rollouts would therefore drive up cost by a factor of $k$. Second, in stateful environments, the environment naturally transitions to the next state after executing an action. Consequently, obtaining multiple rollouts from the same state would require environment resets. In certain scenarios, such as simulations, resetting the environment is possible, but in many scenarios, this is not feasible. Therefore, we rely on standard MC-baselines to estimate $A_{adv}$.

For bandit experiments, we maintain a pool of 512 stochastic MABs. For every rollout, we let the agent interact with a subset of 16 bandits for a single episode (50 timesteps). Consequently, every rollout contains 800 transitions. Similarly, for Tic-tac-toe, we maintain 16 parallel environments and collect 2048 rollout steps. We conduct 1 and 2 update epochs over the rollout buffer for bandits and Tic-tac-toe, respectively.

## B.3  SFT

For our SFT experiments on MABs, we fine-tune on either the expert action or expert rationales produced by UCB. We employ standard SFT training on the SFT datasets described in Appendix A.1.2 using a cross-entropy objective on the target tokens.

## B.4  EXPLORATION MECHANISMS

In Section 8, we compare a variety of classic exploration mechanisms and LLM-specific approaches and study their effects on agent performance on Gaussian MABs with 10 arms. Here, we provide a description for each mechanism.

**Try-all.** The try-all strategy is inspired by UCB, which incorporates an initial phase in which all untried actions are executed. This is because the UCB values for all untried actions are $\infty$. Therefore, for try-all, we incorporate the same exploration phase when performing ICL and RLFT at the beginning of every episode. To enable fine-tuning on exploration actions, we provide an action rationale template to the model (e.g., `Action X has not been tried yet, let's explore it.` `ACTION=X`). While simple, we find that this try-all strategy is effective for lowering regret across all model sizes (see Figure 8). This suggests that the model is able to select appropriate actions if given sufficient information about the consequences of all actions, but struggles to explore actions by itself.

$\epsilon$**-greedy.** $\epsilon$-greedy is classic exploration mechanism and commonly used in RL algorithms (Mnih et al., 2015; Hessel et al., 2018). For our experiments, we use $\epsilon = 0.1$ both during training and evaluation. Note that $\epsilon$-greedy exploration is performed on a per-action level, not on a per-token level. Consequently, at every step, there is a $10\%$ chance that the executed action is sampled uniformly from the action space, instead of being predicted by the agent. In preliminary experiments, we swept over different $\epsilon$ values. However, with $\epsilon$-greedy, we found it difficult to strike a good balance between sufficient action coverage and low regret in our setting. Overall, $\epsilon = 0.1$ enabled stable learning and minor exploration benefits. As for the try-all strategy, we construct a synthetic action rationale template ("Action X has not been tried yet, let's explore it. ACTION=X") to enable RLFT on the random actions.

**Context Randomization.** Context Randomization is an LLM-specific mechanism designed to introduce randomness in the action predictions by modifying the context representation. At every interaction step, we construct a mapping from the original action labels to a shuffled list of the same action labels. Subsequently, we remap action in the context history according to the constructed mapping. Finally, the predicted action is mapped back to the original action label space and executed environment. Besides introducing randomness, context randomization acts as a control mechanism to ensure that the observed biases do not only stem from biases towards particular action-tokens (e.g., blue occurs more often than magenta in the pre-training dataset).

**Context Summary.** Similar to Krishnamurthy et al. (2024) and Nie et al. (2024), we evaluate the effects of providing a context summary to the agent. After the context history, we provide the model with a summary of that history that contains the number of times every action has been selected so far, along with their respective mean rewards.

**Self-Correction.** Inspired by Kumar et al. (2024) and Wang et al. (2025), we employ self-correction to the model's predicted actions. First, we let the model generate its initial rationale and corresponding action prediction. Then we append the generated rationale along with a self-correction message (similar to Kumar et al. (2024)) to the input context, and repeat the action generation. Finally, we extract the action from the final response and execute it in the environment. For RLFT, we only fine-tune on the final response, but retain the initial response along with the self-correction message in the context.

**Self-Consistency.** Instead of generating a single answer, self-consistency (Wang et al., 2022) relies on generating multiple responses. Subsequently, self-consistency employs a majority voting mechanism to determine the final response. For our experiments in Figure 8, we report results for self-consistency with 16 generated responses. Instead of majority voting, we experimented with sampling from the respective response distribution or random mechanisms.

**Exploration Bonus.** Finally, we evaluate a reward shaping mechanism in the form of an exploration bonus. In particular, we give an exploration bonus of +1 if the agent selects an action not yet tried within the respective episode. While simple, we find that the exploration bonus effectively narrows the gap to the UCB expert. This highlights the importance of reward shaping for fine-tuning LLMs in decision-making scenarios.

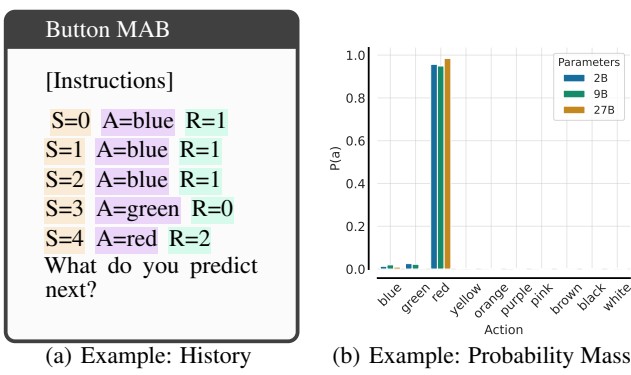

(a) Example: History  (b) Example: Probability Mass

**Figure 15:** Illustration of **action probabilities** leading to greediness behavior. Models exhibit overly high action probabilities in the presence of rewards, potentially resulting in repeatedly selecting sub-optimal actions.

## C ADDITIONAL RESULTS

### C.1 FAILURE MODES

#### C.1.1 GREEDINESS

Greediness is characterized by the LLM overly favoring the best-performing action among a small set of actions seen so far. We define action coverage $C_t$ at step $t$ as the fraction of available actions that have been selected at least once, $C_t = \frac{\{a \in \mathcal{A}: N_t(a) > 0\}}{|\mathcal{A}|}$ with $N_t(a)$ representing the number of times action $a$ has been selected until $t$.

**Action probabilities.** The suboptimal action coverage reported in Section 4.2 is caused by the model overly favoring high-reward actions (i.e., overly high action probabilities). In Figure C.1.1, we provide an illustration of the action probabilities for a given input history. Across model sizes, Gemma2 exhibits overly high action probabilities in the presence of reward, which results in repeatedly selecting a potentially suboptimal action.

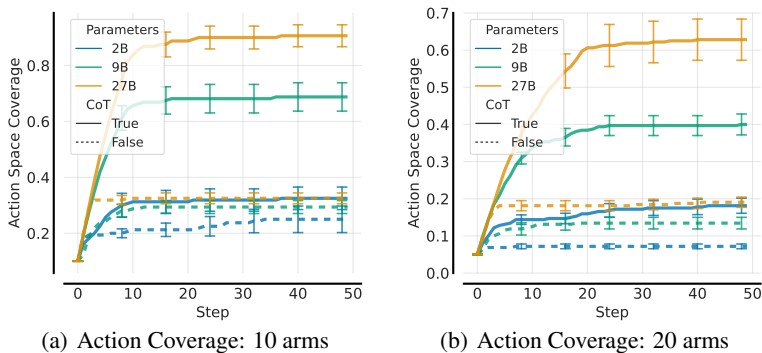

(a) Action Coverage: 10 arms     (b) Action Coverage: 20 arms

**Figure 16:** Illustration of greediness for the **numbers** scenario.

**Greediness on Continuous MABs.** We repeat the analyses conducted in Section 4.2 using *numbers* instead of *buttons* as the possible actions. Indeed, we find that the same trends hold. Without CoT, the performance remains low. For Gemma2 27B, we observe an increase in the action coverage to almost 90% for the 10 arms scenario, and to 60% for the 20 arms scenario.

**Post RLFT.** In line with Figure 17a, we present the post-RLFT action coverage on the 20 arms scenario in Figure 17b. Similar to the effects on the 10 arms scenario, we observe that RLFT improves the action coverage by 13%.

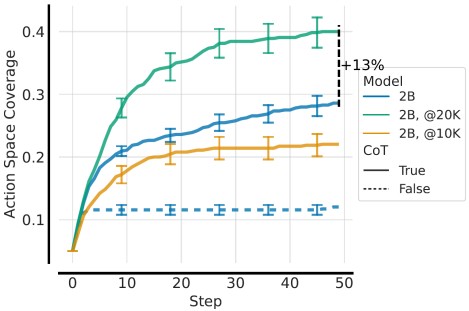

**Figure 17:** Effect of RLFT on **greediness** for 2B on 20 arms (medium noise).

**100-step Horizon.** Throughout our experiments, we limit the horizon to 50 interaction steps (see Figure 3) because of practical reasons. In our setup, the agent has to produce a decision rationale at every step because of the multi-step nature of decision-making. Every step involves both a prefilling stage (encoding instructions, history) and a generation stage for producing the rationale/action, which is costly over multi-turn interactions. Given the flattening of the action coverage curves in Figure 3, we do not expect that the agent would continue to explore over a longer horizon. To verify this

hypothesis, we conduct an additional ablation in which we run the smallest and largest Gemma2 models on the 10 arms scenario with an extended horizon of 100 interaction steps (see Figure 18). Indeed, we find that the action coverage remains flat over the longer horizon, which indicates that the agent does not continue to explore due to greedy action selection strategies.

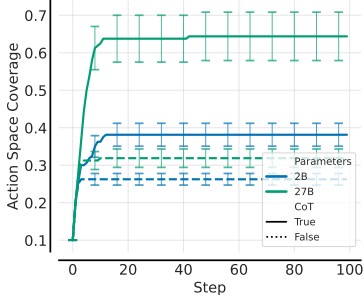

**Figure 18:** Greediness over a longer horizon of 100 interaction steps. We compare Gemma2 2B and 27B with and without CoT. The agents do not continue to explore over longer horizons.

### C.1.2 FREQUENCY BIAS

Frequency bias is characterized by repeatedly selecting the most frequently occurring actions in the context, even when the dominant action gives a low reward. To measure frequency bias, we first construct a variety of interaction histories (occurred during environment interaction) containing between 2 and 10 transitions. This interaction history is collected using a random policy. Given an initial interaction history, we repeat the last action in the history, which we also refer to as the target action, between 0 and 100 times. Finally, we report the entropy all actions, $H(\theta) = -\sum_{a \in A} \pi_\theta(a \mid \tau) \log \pi_\theta(a \mid \tau)$. To achieve this, we conduct a separate forward pass for every possible action in the action space and report the respective log probabilities. We repeat the same procedure for different interaction histories and target actions (see Figure 4a and c). For the 10 arms scenario, every interaction history therefore results in 1000 (10 arms * 100 repetitions of the target action) forward passes. We repeat this procedure for the 5 target actions reported in Figure 4 (i.e., 'green', 'red', 'blue', 'orange', 'black' buttons) using 5 interaction histories per action, accumulating to a total of 25K model forward passes (1000 * 5 * 5) per figure.

To quantify frequency bias, we categorize the resulting actions as *frequent* action, *greedy*, or *other* if they are neither frequent nor greedy. Subsequently, we compute the frequent $F_f$, greedy $F_g$ and other $F_o$ fractions as reported in Figure 4:

$$F_f = \frac{N_T(a_f)}{N}; \quad F_g = \frac{N_T(a_g)}{N}; \quad F_o = \frac{\sum_{a \in A \setminus \{a_f, a_g\}} N_T(a)}{N}, \quad \text{with } N = \sum_{a \in A} N_T(a). \quad (5)$$

Note that there can be an overlap between greedy and frequent actions. In these (rare) cases, the greedy action category is dominant, i.e., we categorize the action as greedy even if it would also be the frequent action. This implies that the actions classified as frequent in Figure 4, are always suboptimal/bad compared to the respective greedy action. Consequently, a high $F_f$ indicates that the model prefers the most frequent action even when observing a better action in the context.

**Post RLFT.** In Section 4.3, we observed that RLFT counteracts frequency bias. In addition to frequency buckets reported in Figure 19a, we plot frequency against action entropy post RLFT in Figure 19b. Compared to Figure 4a, we observe that after RLFT the models maintain a higher action entropy for longer. Only at high repetition frequencies does the action entropy decrease severely. Consequently, RLFT counteracts frequency bias, but does not completely alleviate it.

### C.1.3 KNOWING-DOING GAP

The *knowing-doing gap* has been observed by Paglieri et al. (2024) and Ruoss et al. (2024). It states that models possess knowledge about a task or consequences of their behavior (i.e., they *know* what to do), but cannot materialize this knowledge when acting (i.e., they are incapable of *doing*). However, in these works, it was difficult to characterize the knowing-doing gap empirically, due to the complexity of their environments. In contrast, our simple MAB setting is well-suited to quantify their observations precisely. We illustrate the knowing-doing gap empirically in Figure 5. To this

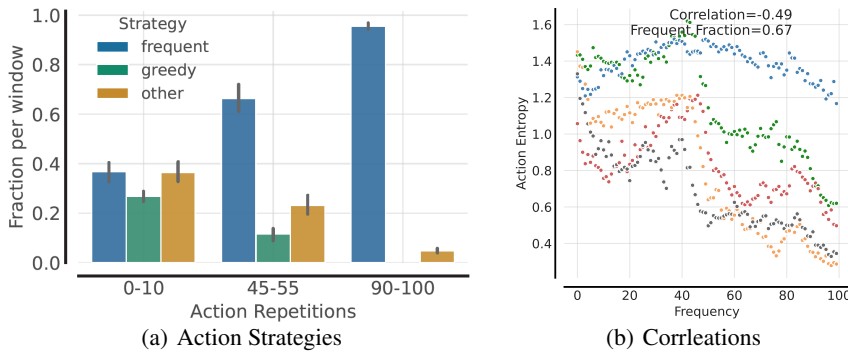

**Figure 19:** Effect of RLFT on **frequency bias** for 2B (10 arms, medium noise).

end, we first task Gemma2 27B to produce the UCB algorithm and to compute the relevant quantities required to act according to UCB ("knowing"). This involves counting how often every action was selected, computing the mean rewards for every action, and computing the final UCB values. After producing the quantities, the model is tasked to act according to them (i.e., "doing"). In Figure 26, we present an example of the final instructions given to the model along with a response produced by Gemma2 27B.

To evaluate performance empirically, we let Gemma2 27B interact with the environment (64 parallel instances) for 50 timesteps. We extend the token generation budget to 2048 tokens per step to accommodate the additional required computations. Every produced action $z$ contains both the CoT rationale $z_{CoT}$ and the final selected action $a$. We first extract the computed UCB values from the produced rationale $z_{CoT}$. To achieve this, we task Gemma2 27B to enclose the computed values by <ucb_values> and </ucb_values> blocks. Then we extract the selected action $a$ and execute it in the environment. For this experiment, we use Gemma2 27B, because we found that 2B and 9B struggled with computing the relevant UCB quantities and with enclosing them appropriately under the desired blocks.

**Quantifying "Knowing".** To quantify "knowing", we compare the UCB values computed by the model and extracted from $z_{CoT}$ against the real UCB values. To this end, we recompute the real UCB values for every action at every time step given the observed history. We consider the rationale as correct if the arm with the highest UCB values matches. We opt for this choice rather than checking for exact equality because we observed that the model struggles with exact calculations for complex operations. This is expected because the necessary computations involve logarithms and square roots of floating-point values. While tool use (e.g., a calculator) could mitigate this issue, we observed that Gemma2 27B gets the quantities approximately right, resulting in valid rationales. Thus, the fraction of correct rationales is $F_c = \frac{1}{N} \sum_{i=1}^{N} g(z_{CoT}^i)$ given a classifier $g$.

**Quantifying "Doing".** To quantify "doing", we categorize the generated actions as *optimal* action if the model selects the action with the highest UCB value, as *greedy* if it selects the action with the highest UCB value among the set of actions tried so far, and as *other* if the action is neither optimal nor greedy. It is possible that the greedy action is the optimal action. However, in this case, the action is considered optimal instead of greedy. Subsequently, we compute the percentages of greedy/optimal/other actions (e.g., $F_g \times 100$). We find that the model clearly *knows* how to solve the task, with 89% of all rationales being correct (see Figure 5).

To ensure that our findings represent a fundamental model bias and not a prompt artifact, we investigated a variety of prompting strategies in preliminary experiments to encourage the LLM to strictly adhere to the UCB algorithm. For example, this involved strong language, including explicit instructions to prioritize untried actions and to prefer actions with high/infinite UCB values. Moreover, we provided hints to the model that actions with infinite UCB are strictly better than other actions and should always be preferred. Across prompting strategies, our findings remained similar.

## C.2 MULTI-ARMED BANDITS

In Figure 6, we report the cumulative regrets across model sizes and arms for a medium noise ($\sigma = 1.0$) scenario. In addition, we repeat the same experiment in the low-noise ($\sigma = 0.1$) and the high-noise $\sigma = 3.0$ setting in Figure 20. For both noise levels, we observe similar trends to those

for the medium noise setting. In particular, we observe that LLMs clearly outperform the random baseline, and RLFT lowers the cumulative regret for Gemma2 2B across all arm scenarios.

We note that in Figure 6, we observed that 9B, ICL outperforms 9B, RLFT. One possible hypothesis for this behavior is that the higher short-term regret may be a direct consequence of RLFT. RLFT encourages the agent to explore more by rewarding reasoning processes that lead to exploratory actions. In this specific scenario, the increased exploration can be costly in the short term over the 50-step horizon, even though it may be essential for long-term performance.

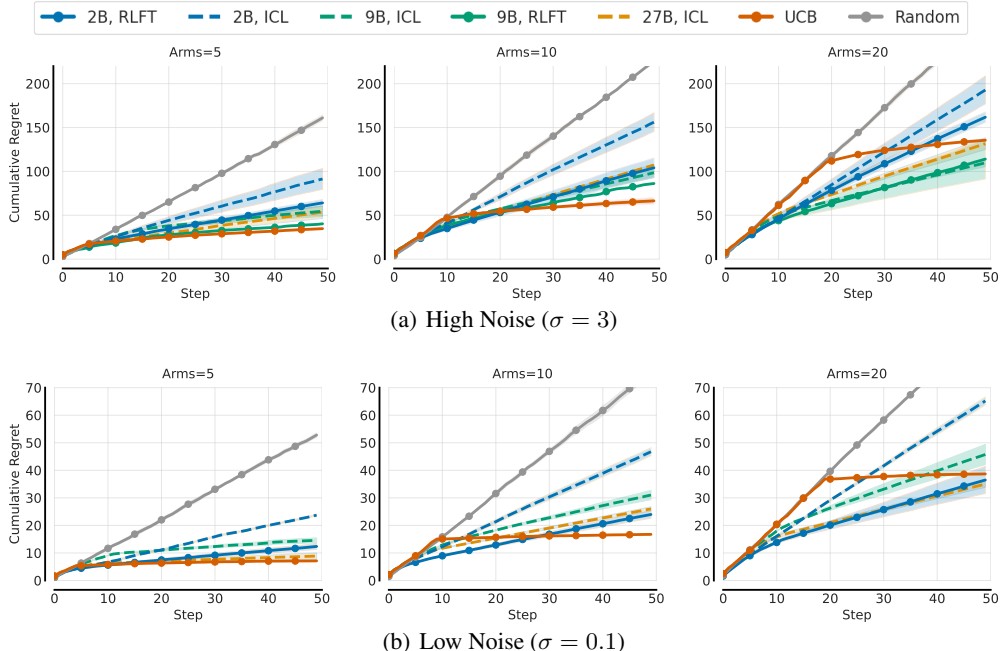

(a) High Noise ($\sigma = 3$)

(b) Low Noise ($\sigma = 0.1$)

**Figure 20:** Main Comparison on **Gaussian MABs** button scenario in the **(a)** high $\sigma = 3$ and **(a)** low $\sigma = 0.1$ noise settings. We compare cumulative regrets (lower is better) of classic baselines against ICL and RLFT performances for Gemma2 2/9/27B for 5, 10, and 20 arms.

### C.3 CONTEXTUAL BANDITS

We repeat the same fine-tuning experiment for the contextual MovieLens bandits described in Section A.2. In Figure 21, we report the cumulative regrets attained by Gemma2B across different model sizes and for 5, 10, and 20 arms. Furthermore, we compare against a LinearUCB and a random baseline. Overall, we observe similar performance improvements for RLFT on CBs as on MABs. While the ICL performances barely attain the same performance as a random agent, RLFT fine-tuned Gemma2 2B performs similarly to UCB.

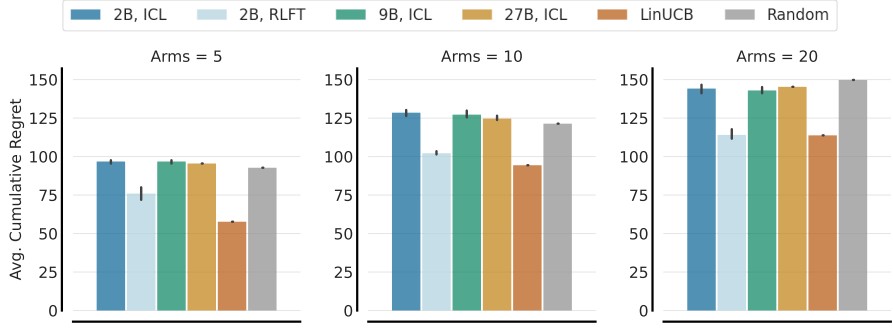

**Figure 21:** Main Comparison on Gaussian **MovieLens CBs** for **(a)** 5, **(b)** 10, and **(c)** 20 arms. We compare classic baselines against ICL and RLFT performances for Gemma2 2/9/27B.

## C.4 OTHER MODEL FAMILIES: QWEN-2.5 AND LLAMA3

To ensure that our results are not specific to the Gemma2 model family, we repeat the greediness analysis reported in Figure 3 with the widely-used instruction-tuned Llama3 (Dubey et al., 2024) and Qwen-2.5 (Qwen et al., 2025) models. We report the action coverages for 10 arms and 20 arms along with the action coverages plotted against cumulative regret for 10 arms in Figure 22 and 23 for Qwen-2.5 and Llama3, respectively.

For Llama3, we evaluate the latest checkpoints for three available model sizes: 3B, 8B, and 70B. Specifically, we use the instruction-tuned variants of Llama 3.2 for 3B, Llama 3.1 for 8B, and Llama 3.3 for 70B. Similarly, we report scores for the instruction-tuned variants of Qwen-2.5 in 4 model sizes: 3B, 7B, 14B, and 32B. We evaluate all checkpoints both with and without CoT instructions using their respective instruction templates.

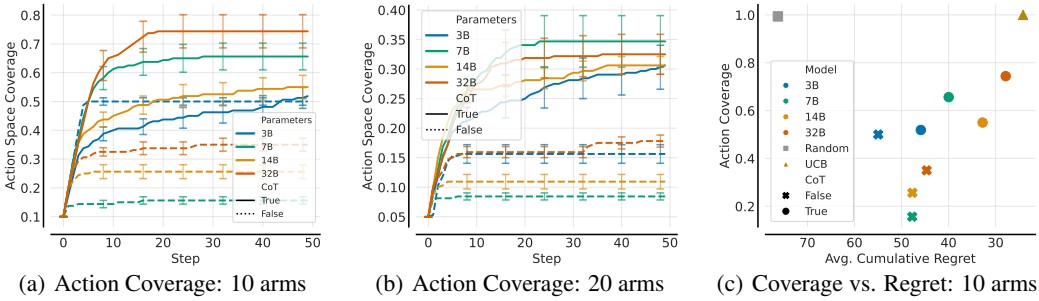

(a) Action Coverage: 10 arms    (b) Action Coverage: 20 arms    (c) Coverage vs. Regret: 10 arms

**Figure 22:** Illustration of **Greediness**. We show action coverage for **Qwen-2.5** 3B/7B/14B/32B instruct models with and without CoT for **(a)** 10 and **(b)** 20 arms over 50 interaction steps. The agents favor the best-performing action among the set of selected actions, leading to stagnating action coverage, despite the benefits of larger models and CoT. In **(c)**, we plot cumulative regret against action coverage. The general trends are similar to Gemma2 and Llama3, suggesting that the limitations persist across model families.

Generally, we observe that Llama3 and Qwen-2.5 models exhibit similar trends in terms of action coverage and regret as Gemma2 models (see Figures 22 and 23). For both model families, CoT considerably improves action coverage and therefore regret. One exception is Qwen-2.5 3B w/o CoT in the 10 arm scenario, which always first selects 5 actions before committing to a particular action. Overall, the smaller-scale Llama3 models (3B, 8B) tend to achieve lower action coverages compared to the Gemma2 and Qwen-2.5 models of similar sizes, both in the 10 and 20-arms scenario. Llama3 70B achieves the highest action coverage across all model sizes, potentially highlighting the benefits of the larger model size. However, in the 20 arms scenario, we still observe a considerable gap to full action coverage for both model families, with Qwen-2.5 and Llama3 selecting 35% and 65% of all actions at maximum, respectively.

We note that for Qwen-2.5, the larger models do not always outperform the smaller models. For example, in 7B attains 10$ higher action coverage than 14B in the 10 arms scenario. This is an interesting observation of non-monotonic scaling, where performance on certain tasks does not strictly increase with model size. We believe that this may stem from differences in the pre-training corpora, token budgets, or alignment techniques applied to specific model checkpoints. We leave a deeper analysis of this phenomenon for future work. Across model sizes, it is, however, apparent that the action coverage flattens out over the 50-step horizon. This indicates that all model sizes suffer from the same fundamental bias.

## C.5 PERCENTAGE OF INVALID ACTIONS.

In the case that no valid action is extracted from the decision rationale, a random action is executed. This mechanism serves as a safeguard to ensure that the environment interaction continues without interruptions. To verify that the safeguard does not bias our experimental results, we report the percentage of invalid and consequently random actions performed across the 50-step horizon for all model families in Table 2.

For the experiments reported in Figures 3, 22, and 23, we recorded whether the executed action was valid or invalid. To compute the percentages reported in Table 2, we first calculate the fraction of random actions per episode (i.e., over the 50-step horizon) and subsequently average over the 64 bandits to get a more robust statistical estimate. Indeed, we find that only a small percentage of all

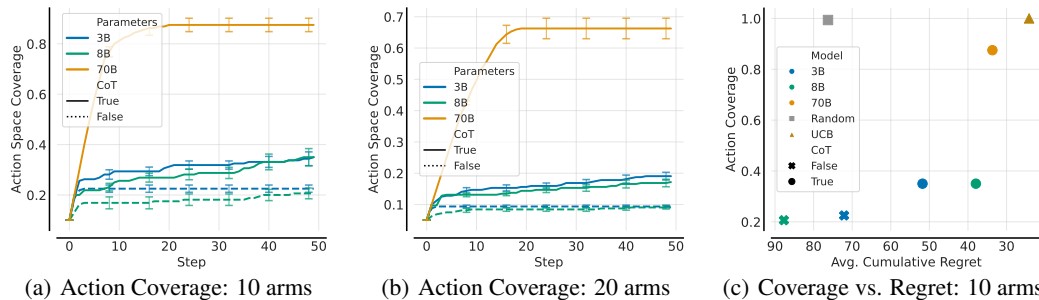

(a) Action Coverage: 10 arms    (b) Action Coverage: 20 arms    (c) Coverage vs. Regret: 10 arms

**Figure 23:** Illustration of **Greediness**. We show action coverage for **Llama3** 3B/8B/70B instruct models with and without CoT for **(a)** 10 and **(b)** 20 arms over 50 interaction steps. The agents favor the best-performing action among the set of selected actions, leading to stagnating action coverage, despite the benefits of larger models and CoT. In **(c)**, we plot cumulative regret against action coverage. We use Llama 3.2, 3.1, and 3.3 for 3B, 8B, and 70B, respectively. Note that we do not report results for Llama3 70B w/o CoT, as we did not get the model to produce valid actions without CoT instructions. The general trends are similar to Gemma2 and Qwen-2.5, suggesting that the limitations persist across model families.

actions is invalid. For example, 2% of the 50 interaction steps indicates that, on average, only a single action is invalid per episode. Consequently, the safeguard does not introduce any meaningful bias into our results. Interestingly, we found that for many trials, the very first interaction in the trial resulted in an invalid action, with the model often producing `ACTION=random` as its final output. We assume that this is due to the empty interaction history in the first step.

**Table 2:** Percentages of invalid/random actions performed over the 50-step horizon on the Gaussian button bandit with 10 arms for all model families.

| Model | Chain of Thought | % of invalid/random actions |
|---|---|---|
| Gemma2 2B | ✕ | 2.00 |
| Gemma2 2B | ✓ | 3.19 |
| Gemma2 9B | ✕ | 2.00 |
| Gemma2 9B | ✓ | 2.00 |
| Gemma2 27B | ✕ | 2.00 |
| Gemma2 27B | ✓ | 0.00 |
| Llama3 3B | ✕ | 2.00 |
| Llama3 3B | ✓ | 1.50 |
| Llama3 8B | ✕ | 2.00 |
| Llama3 8B | ✓ | 1.50 |
| Llama3 70B | ✓ | 0.00 |
| Qwen 2.5 3B | ✕ | 0.00 |
| Qwen 2.5 3B | ✓ | 2.00 |
| Qwen 2.5 8B | ✕ | 0.00 |
| Qwen 2.5 8B | ✓ | 0.00 |
| Qwen 2.5 14B | ✕ | 2.00 |
| Qwen 2.5 14B | ✓ | 0.00 |
| Qwen 2.5 32B | ✕ | 0.25 |
| Qwen 2.5 32B | ✓ | 0.00 |

### C.6 PROMPT TEMPLATE VARIATIONS.

To better understand the influence of the position of the interaction history on final performance, we conduct an additional ablation study using Gemma2 2B with different prompt templates. In our setup, the input context contains 3 sections, as illustrated in Figure 1: (1) task instructions, (2) output instructions, and (3) the interaction history. Accordingly, we mix up the order of these sections, resulting in 3 context variations: `task-output-history`, `task-history-output`, `history-task-output`.

Across prompt templates, we observe similar patterns with action coverages flattening out over the 50-step horizon (see Figure 24). Nevertheless, some differences exist among the templates.

For example, `history-task-output` performs worst overall without CoT. Overall, positioning the interaction history last (our default setting) performed best in the CoT setting.

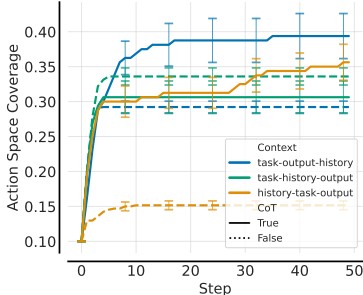

**Figure 24:** Illustration of the effect of different prompt templates. We compare Gemma2 2B with and without CoT using three different context orderings: `task-output-history`, `task-history-output`, `history-task-output`. While differences exist, the variations exhibit similar patterns with action coverages flattening out after a handful of steps.

### C.7  EFFECT OF EXPLORATION MECHANISMS

For RLFT, we relied solely on the exploration properties for CoT reasoning. Therefore, in Section 4.4 we studied the effects of classic exploration mechanisms and LLM-specific strategies to encourage exploration. In particular, we compare: **(1)** try-all actions initially similar to UCB, **(2)** $\epsilon$-greedy, **(3)** context randomization, **(4)** context summary similar to Krishnamurthy et al. (2024) and Nie et al. (2024), **(5)** self-correction similar to Kumar et al. (2024), **(6)** self-consistency (Wang et al., 2022), and **(7)** exploration bonus.

In Figure 8, we observe that the mechanisms result in varied effects on action coverage First, we find that the simple *try-all* strategy, which reduces the need for additional exploration by trying all actions, results in the biggest performance improvements. This suggests that given sufficient information about the (sub-)optimality of actions, LLMs are better able to select actions accordingly, underscoring their exploration shortcomings. Second, a simple *exploration bonus* (+1 reward for untried actions during RLFT) significantly increases exploration ($50\% \rightarrow 70\%$) and lowers regret towards the expert compared to regular RLFT. This highlights the importance of reward shaping for fine-tuning LLMs to elucidate a desired behavior.

## D  ABLATIONS

### D.1  RLFT IN TIC-TAC-TOE.

To investigate the efficacy of RLFT in stateful environments, we evaluate on Tic-tac-toe from Ruoss et al. (2024), in which frontier models struggle to achieve strong performance (see Appendix B for training details). We fine-tune against three opponents: a random agent, Monte Carlo Tree Search (MCTS) (Coulom, 2006), and noisy MCTS (50% of actions selected at random). We find that RLFT significantly enhances the win-rate of Gemma2 2B against all opponents compared to ICL (see Figure 9a). Against the random agent, RLFT elevates the average return from 0.15 (i.e., winning 15% of games) to 0.75. Notably, the agent even manages to draw against the optimal MCTS baseline ($-0.95 \rightarrow 0.0$), underscoring the effectiveness of RLFT for decision-making. However, for high performance, it is essential to provide the legal actions in the context (see Figure 25).

### D.2  TIC-TAC-TOE: EFFECT OF LEGAL ACTIONS IN STATE

By default, we provided the legal actions available at the current turn within the input context to the agent. We found this design choice to be essential for effective fine-tuning compared to training without legal actions (see Figure 9b). Without legal actions in the context, the average return drops from 0.75 (w/ legal actions) to 0.45. This suggests that the LLM fails at identifying the appropriate actions among the set of all possible actions when not given legal actions at the current state. In contrast, when provided with sufficient information (i.e., legal actions), the LLM is able to select actions appropriately (similar to Section 4.4). Providing the legal actions in the agent's

context alleviates the need to explore/identify invalid actions. Consequently, this shortcoming further highlights the need for principled exploration strategies for LLMs in decision-making scenarios.

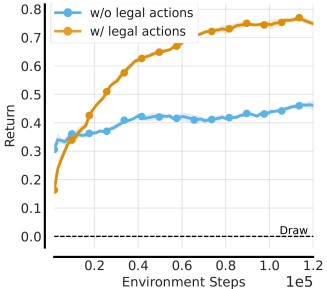

**Figure 25:** Effect of removing **legal actions** from the context in Tic-tac-toe.

### D.3 IMPORTANCE OF CoT FOR RLFT

CoT reasoning is critical for ICL performance (see Figure 3), but the question remains how CoT influences RLFT. Therefore, we run RLFT on Gemma2 2B on the 10 arms Gaussian MAB both with and without CoT (see Figure 9b, RLFT). Indeed, without CoT, RLFT barely attains the performance of ICL w/ CoT. This highlights the function of CoT as a vital exploration and rationalization mechanism for decision-making. For our results without CoT reported in Figure 9b, we remove the CoT instructions given to our agents. Instead, we instruct the agents not to perform any reasoning steps and to only produce the action to execute in the environment $a$. We limit the token generation budget $G$ to 16 to avoid the model ignoring the instructions. Furthermore, this considerably speeds up training due to faster rollout times and shorter context lengths.

### D.4 EXPERT BEHAVIOR CLONING VS. THOUGHT CLONING

A prevalent approach in sequence models for decision-making is behavior cloning (BC) (Pomerleau, 1988; Reed et al., 2022; Brohan et al., 2022; 2023), which relies on expert datasets. Consequently, we construct two UCB expert datasets comprising 32K rollouts either w/o CoT (*behavior cloning*) or w/ CoT (*thought cloning*), as described in Appendix A.1. Notably, both SFT variants successfully mimic the expert, achieving comparable regret to the UCB expert (see Figure 9b, SFT). This result underscores the efficacy of training on expert data in decision-making scenarios when available, echoing recent findings in reasoning tasks (Muennighoff et al., 2025). While BC and TC attain similar performance levels on the simplistic MABs, we anticipate that TC is advantageous in more complex decision-making scenarios as found by Hu & Clune (2023).

### D.5 "THINKING" TIME

We investigate the effect of giving the agent more time to "think" in Figure 9c. To achieve this, we vary the maximal number of tokens that the agent can generate per action $G \in \{16, 64, 256, 512\}$. By default, we set $G$ to 256. Indeed, we observe that the performance improves consistently with more thinking tokens. Decreasing $G$ to 16 or 64 results in poor performance, because the agent is unable to rationalize its decisions within the restricted generation budget. This is similar to the performance without CoT, but in contrast, the agent is instructed to produce the reasoning process. Over the course of RLFT, the agents learn to produce short rationales $z_{CoT}$, including the action $a$ due to our reward shaping mechanism (see Section 3.2). However, the produced short rationales are unhelpful to improving agent performance.

In contrast, doubling $G$ from 256 to 512 results in a considerable performance increase to the level of Gemma2 9B with RLFT (see Figure 6). We observe an increase in the average sequence length over the course of the RLFT process. This suggests that the agent learns to effectively leverage the additional "thinking time" and reflects recent observations in mathematical reasoning (Guo et al., 2025). However, the increased performance comes with additional training cost due to the multistep nature of decision-making scenarios. In fact, we observed that rollout generation can make up the majority of the training time required by the RLFT process. This is because the agent has to produce more tokens at every environment interaction step. For example, for our default horizon of 50 timesteps and a generation budget of 500, the agent produces 25K tokens (at maximum).

---

**Instructions for Gemma2 27B as UCB agent**

[Instructions]

Your task is to act according to the Upper-Confidence-Bound (UCB) algorithm. First, write down the UCB algorithm. Then compute the relevant UCB values for every button (approximate if necessary). Finally, select your action according to the computed quantities.

[More Instructions]

So far you have tried/seen:
Step=0 Action=blue Reward=1.06
Step=1 Action=blue Reward=1.82
Step=2 Action=green Reward=1.0
Step=3 Action=green Reward=-0.26
Step=4 Action=blue Reward=-0.58
Step=5 Action=blue Reward=-0.34
Step=6 Action=green Reward=1.19
Step=7 Action=green Reward=2.21
Step=8 Action=green Reward=0.07
Step=9 Action=green Reward=1.45
Step=10 Action=green Reward=-0.11
What do you predict next?

---

**Generated response by Gemma2 27B**

Here's how I'd approach this as a UCB bandit algorithm:

**UCB Algorithm (Short Version)**

- **Initialization:** For each button, set a count (number of times pressed) to 0 and an estimated reward average to 0.

- **Iteration:**

  - For each button, calculate the UCB value: $UCB = \text{average\_reward} + \sqrt{\frac{2 \cdot \ln(\text{total\_steps})}{\text{button\_count}}}$

  - Choose the button with the highest UCB value.

  - Update the button's count and average reward based on the observed reward.

**UCB Calculations (Approximate)**

Let's denote the buttons by their colors. Here's a breakdown after 10 steps:

- **blue:**
  - Count = 4
  - Average Reward = (1.06 + 1.82 - 0.58 - 0.34) / 4 = 0.49
  - UCB $\approx 0.49 + \sqrt{\frac{2 \cdot \ln(11)}{4}} \approx$ 1.17

- **green:**
  - Count = 7
  - Average Reward = (1.0 + -0.26 + 1.19 + 2.21 + 0.07 + 1.45 - 0.11) / 7 $\approx$ 0.92
  - UCB $\approx 0.92 + \sqrt{\frac{2 \cdot \ln(11)}{7}} \approx$ 1.47

- **Other buttons:** Count = 0, Average Reward = 0, UCB = $\sqrt{\frac{2 \cdot \ln(11)}{0}} = \infty$ (We'll explore these later)

**Action Selection:** The button with the highest UCB value is **green**.
**Therefore, ACTION=green**
**Explanation:** The UCB algorithm balances exploration (trying buttons we haven't seen much) and exploitation (choosing buttons that seem to have high rewards).

- Green has a higher average reward than blue, but blue has been explored less.

- The UCB formula takes this into account, giving a higher value to green because its average reward is more reliable due to more observations.

- The other buttons have infinite UCB values because we haven't tried them yet, so they are highly prioritized for exploration. However, green's current UCB is high enough to warrant another try.

**Figure 26:** Illustration of the **knowing-doing gap**. **(a)** Instructions for the agent, which is prompted to act like a UCB algorithm. **(b)** The response generated by Gemma2 27B with greedy decoding (temperature=0). The LLM "knows" the UCB algorithm and computes the relevant quantities approximately correctly, but acts erroneously by selecting the next action greedily.

