# OpenReview forum: "LLMs are Greedy Agents: Effects of RL Fine-tuning on Decision-Making Abilities"
_ICLR.cc/2026/Conference — ICLR 2026 Poster_

### Official Review · Reviewer_8Kgn · 2025-10-23

**Soundness:** 2
**Presentation:** 2
**Contribution:** 2
**Rating:** 4
**Confidence:** 3

**Summary:**

This paper investigates the decision-making capabilities of LLMs under uncertainty using multi-armed and contextual bandit settings. Through analyses including action coverage tracking and comparison against optimal algorithms like UCB, the authors identify three primary failure modes: greediness, frequency bias, and the knowing-doing gap.

To mitigate these, the paper fine-tunes Gemma2 models (2B, 9B, 27B) using PPO trained with CoT reasoning traces on environment rewards. The PPO method significantly improves performance over the baseline ICL methods. The study further shows that augmenting PPO with classic exploration techniques (like try-all, ε-greedy, and exploration bonuses) yields additional gains, with methods like adding an exploration bonus or an initial "try-all" phase proving most effective among the tested PPO variations, and provided ablation results on CoT, thinking time, and additional experiments on Tic-tac-toe.

**Strengths:**

The paper provides a valuable diagnostic analysis of why LLMs fail in decision-making tasks, systematically identifying and quantifying specific issues like greediness, frequency bias, and the knowing-doing gap using bandit environments. This clear characterization of common failure patterns is a useful contribution.
The work demonstrates that PPO fine-tuning with CoT traces effectively improves LLM decision-making across various tasks (MAB, CB, Tic-tac-toe) and model scales, outperforming baseline in-context learning without requiring expert data. Furthermore, the study evaluates integrating simple exploration strategies during PPO, showing that techniques implemented via prompt/context modifications, such as initializing with a "try-all" phase, yield additional performance gains. Overall the paper offers novel insights into practical methods for enhancing exploration of PPO-tuned LLM agents under related decision making settings.

**Weaknesses:**

Limited Generalizability: The paper claims to analyze general LLM agent behavior but bases its core analysis almost entirely on simple bandit tasks (MAB and CB). The setting is somewhat narrow (also, SOTA/commercial models were not evaluated), making it hard to generalize the findings to more complex agentic situations. The knowing-doing gap analysis is a particularly interesting topic, but the analysis is tied to the UCB structure in bandits; it would be interesting to explore a more generic analysis that can characterize the knowing-doing gap across a variety of decision-making tasks (possibly with some form of optimal algorithm as part of the LLM’s internal knowledge) and provide deeper insights (e.g., could this be a value alignment problem where the LLM’s default option is not using an optimal algorithm?).

Method Performance and Novelty: While the proposed PPO fine-tuning method improves LLM performance over the ICL baseline, it doesn't match the results of SFT using expert UCB data. Given that SFT data is relatively easy to synthesize for MAB tasks, the proposed method is less useful in this context. The PPO solution itself, while possibly the first published work on MAB/CB, is a well-established method, making it less novel. The authors are encouraged to expand the method to more decision-making settings (e.g., more text-games as in https://arxiv.org/abs/2502.17543) and showcase more solid results on improving exploration there.

Finally, the Tic-tac-toe game feels less like an ablation to me. It would be interesting to have full coverage of the analysis there.

**Questions:**

Have we explored prompting methods to ensure LLM strictly follows UCB algorithm?

For the 𝜖-greedy method: is it done at token level or at action level? If it’s the latter, how is it implemented?

---

> ### Author Response · Authors · 2025-11-21
>
> Thank you for your helpful feedback. We are happy to hear that you believe that our work provides a valuable diagnostic analysis and offers novel insights into LLMs in decision-making scenarios. We address your open points below.
>
> **Generalizability to complex agentic situations:**
>
> * We understand the reviewers' interest in studying more complex tasks. In this work, however, we intentionally focused on simple environments (MABs, tic-tac-toe). This simple setup allows us to study the decision-making abilities of LLMs in a principled way, without the complications of complex environments. Our in-depth analysis aims to provide valuable insights for future works, as noted by the reviewer. We agree that studying the effects of RLFT in more complex environments in which exploration is critical will be valuable. Therefore, we have listed this point in our future work section.
> * Even in our restricted setting, we found that small-to-medium scale LLMs (Gemma2 2/9/27B, Qwen 2.5 3/7/14/32B, and Llama 3 3/8/70B) often behave suboptimally, as characterized by the three failure modes we analyzed. Therefore, we expect that they may exhibit related biases in more complex agentic scenarios, which is also indicated by two recent benchmarks for agentic LLMs [1,2]. For example, in [2], frontier models struggle in more complex environments with high-dimensional state spaces, such as Crafter. Nevertheless, we did observe positive effects of RLFT on the agentic abilities of LLMs in MABs/Tic-tac-toe, and we expect that RLFT will also help in more complex environments.
> * Thank you for your suggestion to study the knowing-doing gap across more decision-making tasks. We deliberately chose simple environments to study the knowing-doing gap, as the restricted setup allows for a rigorous empirical quantification of the gap. More complex scenarios would require careful experiment design to enable a robust empirical quantification. While we cannot provide such an extended study during the rebuttal period, we agree that this is a very fruitful avenue for future work. Environments from [3], as referenced by the reviewer, could be a great target for studying the knowing-doing gap further. We added this avenue to our updated manuscript.
>
> **Method performance:**
>
> * We want to clarify that the goal of this study is not to outperform established bandit algorithms (e.g., UCB) using LLMs and RLFT in terms of regret. Instead, we consider UCB an upper bound (see Section 4.1). Our work focuses on analyzing how LLMs behave in simple decision-making scenarios and aims to contribute to a better understanding of potential shortcomings.
> * In particular, we study what effects our approach to RLFT has on the decision-making abilities of LLMs and how it influences the identified shortcomings. We find positive effects, as outlined in the manuscript (e.g., by increasing action coverage and decreasing frequency bias). While RLFT alleviates the studied failure modes, it does not completely solve them.
> * Similarly, our goal is not to outperform SFT on expert data (see Figure 9b). The expert data is directly obtained by creating synthetic language traces using UCB behavior. Therefore, we did not expect to outperform an agent fine-tuned on expert data, which encodes the UCB behavior. Instead, this ablation indicates the effectiveness of using expert data when available.
>
> **Methodological novelty:**
>
> The primary contribution of this paper is to study how LLMs behave in simple decision-making scenarios and to analyze when and why they may behave suboptimally. We use RLFT as a scientific tool to causally probe and mitigate the identified failure modes in these scenarios. Consequently, our work is not meant to be a method paper, but instead focuses on analysis, intended to help us understand potential issues in using LLMs for decision-making. As correctly noted by the reviewer, we are not the first to introduce RL fine-tuning, but rely on established techniques, such as PPO [4]. While we do not aim to propose a new method, we make adaptations to existing approaches to make them suitable for our LLM setting. These changes required careful consideration, and we found them critical for effective RLFT (e.g., reward shaping and action extraction, see Section 3.2).
>
> **Tic-tac-toe analyses:**
>
> We agree that the current presentation of the tic-tac-toe experiments takes the form of an ablation on the effect of RLFT in stateful environments. However, Tic-tac-toe served as a motivating example for us to better understand the agentic abilities of LLM agents in decision-making scenarios. In fact, in [2], the authors showed that frontier models struggle when playing Tic-tac-toe against a random agent. In our experiments, we found meaningful positive effects of RLFT even in Tic-tac-toe.

---

> > ### Author Response · Authors · 2025-11-21
> >
> > Regarding your additional **questions**:
> >
> > * Yes, to ensure that our findings represent a fundamental model bias and not a prompt artifact, we investigated a variety of prompting strategies to encourage the LLM to strictly adhere to the UCB algorithm in preliminary experiments for the knowing-doing gap experiment. For example, this involved strong language, including explicit instructions to prioritize untried actions or to prefer actions with high/infinite UCB values. However, across prompting strategies, our findings remained similar. We added a paragraph on our experience to Appendix C.1.3.
> > * The $\epsilon$-greedy exploration is performed on an action-level. At every step, there is a 10% probability ($\epsilon=0.1$) that the executed action is sampled uniformly from the action space, instead of being predicted by the agent (also see Appendix B.4). To enable RLFT on these actions, we construct a synthetic action rationale (“Action X has not been tried yet, let’s explore it. ACTION=X”) using that action. In preliminary experiments, we swept over different $\epsilon$ values. However, with $\epsilon$-greedy, we found it difficult to strike a good balance between sufficient action coverage and low regret in our setting. We recognize that this was not explained in detail in our manuscript and therefore added a paragraph to Appendix B.4.
> >
> > Thank you again for your comments. We hope our response clarified your questions and the positioning of our paper. If so, we would appreciate it if you considered raising your score. If any questions remain or come up during the discussion period, we would be happy to engage further.
> >
> > [1] Paglieri et al., “Balrog: Benchmarking Agentic LLM and VLM Reasoning on Games”, 2025\
> > [2] Ruoss et al., “LMAct: A Benchmark for In-Context Imitation Learning with Long Multimodal Demonstrations”, 2024\
> > [3] Tajwar et al., “Training a Generally Curious Agent”, 2025\
> > [4] J. Schulman et al., “Proximal Policy Optimization Algorithms”, 2017

---

> > > ### Comment · Reviewer_8Kgn · 2025-11-27
> > >
> > > Thank you for your detailed response. I acknowledge that the paper offers interesting findings regarding the three failure modes, which stands as its primary contribution, a point also highlighted by Reviewer ytZh. I appreciate the clarification that this work is intended to be diagnostic rather than a method paper proposing a novel algorithm.
> > >
> > > However, I maintain my concern regarding the generalizability of these findings to other tasks and larger models. The contribution remains somewhat limited if the findings are primarily demonstrated through MAB. Key questions remain regarding the broader applicability of this diagnosis: Do these findings generalize to more complex decision-making tasks? Do the failure modes persist for significantly larger models, or does scaling generally improve performance across these decision-making structures? If scaling is insufficient, are there any novel techniques?
> > >
> > > Further exploration in these categories would significantly help improve the contribution. Consequently, I will keep my original score.

---

> > > > ### Author Response · Authors · 2025-12-03
> > > >
> > > > We thank the reviewer for their response. We are happy that you appreciate our clarification on the positioning of our work.
> > > >
> > > > We understand the reviewer's remaining concern about larger models and more complex tasks. We agree that this is an important direction for future work. In this work, we aimed to make progress on understanding the decision-making abilities of LLMs in a controlled and accessible setting. We believe that our analysis already provides valuable insights that are informative beyond the specific tasks we consider.
> > > >
> > > > Regarding the **model scale**, our manuscript includes experiments up to 27B/70B models. Across model scales, we find qualitatively similar behaviors, even though the exact performance levels differ. This suggests that the shortcomings we observed are not fully resolved by scale alone. While we agree that an investigation into frontier models and larger models beyond 27B/70B is an important direction, we put a deliberate focus on these more accessible small-to-medium-scale models for two reasons. First, the multi-step nature of decision-making renders rollout generation with LLMs very costly. Focusing on small-to-medium scale models enables us to systematically probe the decision-making abilities of LLMs in a broader range of settings and to perform the in-depth analysis presented here. Second, large-to-medium-scale models are widely used in practice. Therefore, understanding and mitigating the shortcomings of these models is valuable in its own right.
> > > >
> > > > Regarding the **task complexity**, we understand the reviewer’s interest in studying more complex tasks. In this work, we intentionally focused on simple environments (MABs, tic-tac-toe). This simple setup allows us to study the decision-making abilities of LLMs in a principled way, without the complications introduced by complex environments. Moreover, conducting multi-step rollouts in complex environments, which usually involve long horizons, would further exacerbate the computational burden discussed above, making it difficult to perform the systematic analysis we present in this work. Across these simple settings we considered, we observed similar behavioral patterns across model sizes and families. This consistency suggests that the patterns we identified are not tied to simple environments, but may transfer to more complex environments, even if they manifest differently in such environments. As noted by the reviewer, we see our in-depth analysis in simple environments as providing valuable foundational insights for future works to build upon.
> > > >
> > > > We hope this further clarifies the positioning of our work. We will clarify the intended scope of our work more explicitly in our final version and look forward to studying larger models in more complex decision-making scenarios in future research.

---

### Official Review · Reviewer_HhsF · 2025-10-29

**Soundness:** 3
**Presentation:** 4
**Contribution:** 3
**Rating:** 8
**Confidence:** 4

**Summary:**

This paper presents a systematic investigation into the causes of suboptimal performance in Large Language Model (LLM) agents on decision-making tasks. The authors convincingly identify and quantify three prevalent failure modes: greediness, frequency bias, and the knowing-doing gap. To address these issues, the paper proposes a method of Reinforcement Learning Fine-Tuning (RLFT) on the model's self-generated Chains of Thought (CoT). Through meticulously designed experiments in environments such as multi-armed bandits, contextual bandits, and text-based Tic-Tac-Toe, the paper demonstrates that RLFT effectively enhances the model's exploration capabilities and bridges the knowing-doing gap, thereby improving the overall quality of the LLM's decisions. The study is further enriched by an in-depth analysis of various classical and LLM-specific exploration mechanisms, along with insightful ablation studies.

**Strengths:**

- A key strength of this paper is its thorough and systematic diagnosis of the problem before proposing a solution. The clear definition and empirical quantification of greediness, frequency bias, and the knowing-doing gap provide a valuable conceptual framework and a strong motivation for the proposed method.
- The authors evaluate their approach across multiple model scales (Gemma2 2B, 9B, 27B) and environments of varying complexity. The inclusion of strong baselines like UCB and extensive ablation studies (e.g., on the importance of CoT, "thinking time") significantly strengthens the credibility of the findings. The analysis is also extended to other model families (Llama3, Qwen2.5) in the appendix, further demonstrating the generality of the results.
- The proposed RLFT on self-generated CoT is a logical and effective approach. By directly acting on the model's internal reasoning process, it offers a more fundamental way to improve external behavior compared to fine-tuning on actions alone. The experimental results clearly demonstrate the method's positive impact on increasing action coverage and reducing cumulative regret, directly linking the solution to the identified problems.

**Weaknesses:**

- The paper effectively demonstrates that "external" exploration scaffolds (e.g., the try-all strategy or exploration bonuses) improve performance. This suggests that the model may still lack a fundamental, intrinsic drive for exploration. A valuable direction for future work could be to explore integrating intrinsic motivation directly into the RLFT objective. For instance, adding an entropy bonus on the action distribution or novelty-based rewards (e.g., based on state visitation counts) might encourage the model to develop a more generalizable and autonomous exploration policy.
- The paper employs a regex-based method for action extraction and applies a fixed penalty (-5) for invalid outputs. While practical, the robustness of this approach could be further discussed. A comparison with increasingly standard structured output formats (e.g., JSON objects) would be a valuable addition. Furthermore, a sensitivity analysis on the magnitude of the penalty for invalid actions would offer valuable insights into its impact on the learning dynamics.

**Questions:**

- Regarding Figure 1, how significant is the position of the "interaction history" within the prompt on the LLM's performance? A robustness and sensitivity analysis of the prompt template would be highly beneficial for the community.
- The paper shows that explicit exploration rewards are highly effective. As a follow-up, have the authors considered integrating intrinsic exploration motives (e.g., using an entropy bonus or curiosity-driven rewards) directly into the RLFT objective? A comparison with intrinsic exploration mechanisms would further enhance the paper's comprehensiveness.
- On line 192, you mention: "Instead of exploiting multiple rollouts, as used by Ahmadian et al. (2024) and Ramesh et al. (2024), we compute rewards-to-go." Could you please elaborate on the reasoning for this choice, perhaps in the appendix?

---

> ### Author Response · Authors · 2025-11-21
>
> Thank you for your helpful feedback and positive assessment of our work. We are glad that you value the thorough diagnostic nature of our work and find our evaluations across model scales/families useful.
>
> **Intrinsic drive for exploration:**
>
> We strongly agree with the reviewer that studying intrinsic exploration mechanisms is a valuable direction for future work. A variety of intrinsic exploration mechanisms have been proposed in the RL literature, such as surprise-based [1,2] and count-based strategies [3,4], which may transfer well to agentic LLM settings. In fact, [5] have recently leveraged RND [2] in their RL pipeline and found positive effects in single-turn math problems. While we haven’t experimented with intrinsic exploration mechanisms in our current work, we believe they are a promising direction in decision-making scenarios and multi-turn applications that require acting over long horizons (e.g., coding agents). We added a point on this to our future work section.
>
> **Design Decisions:**
>
> * As correctly noted by the reviewer, we make use of a stack of regex expressions to extract the action to execute from the generated tokens (see Appendix B.2). We opted for a flexible regex format (ACTION=X) over more right structured outputs because our preliminary experiments showed it was more robust to varied, sometimes malformed CoT outputs, especially during early RLFT stages. This design choice prioritized more reliable signal extraction during training over strict format adherence. However, depending on the model choice, it makes sense to consider different established extraction approaches (e.g., \boxed{}). We added a note on this rationale to Appendix B.2.
> * We indeed applied a reward penalty of -5 for invalid outputs. To ensure that the reward penalty does not overly bias the optimization, we employ reward normalization to the environment rewards. In preliminary experiments, we found that a range of values between [-10, -1] resulted in stable training progress. In contrast, a high penalty of -100 caused instability. While we cannot provide a sensitivity analysis during the timeframe of the rebuttal, we added a paragraph on our findings to Appendix B.2.

---

> > ### Author Response · Authors · 2025-11-21
> >
> > Regarding your additional **questions**:
> >
> > * To better understand the influence of the position of the interaction history on final performance, we conduct an additional ablation study using Gemma2 2B with different prompt templates in Figure 24 of our updated manuscript. In particular, we vary the position of the interaction history, as suggested by the reviewer. The input context contains 3 sections as illustrated in Figure 1: task instructions, output instructions, and the interaction history. Accordingly, we mix up the order of these sections, resulting in 3 context variations: task-output-history, task-history-output, history-task-output. Across prompt templates, we observe similar patterns with action coverages flattening out over the 50-step horizon. Nevertheless, minor differences exist among the templates. Overall, positioning the interaction history last (our default setting) performed best in the CoT setting.
> > * As mentioned above, we haven’t experimented with intrinsic rewards in our current work, but believe them to be a very fruitful direction for future work. Intrinsic rewards can not only be useful for in-context decision-making, but also for steering the RLFT process. We have experimented with incorporating an entropy bonus into the RLFT process on a token-level, but found this approach tricky to optimize.
> > * Thank you for the question. We expanded Appendix B.2 to discuss this design decision in more detail. We opt for this choice because obtaining multiple rollouts from the same state, as are required for GRPO [6] and RLOO [7], is difficult in decision-making scenarios for two reasons. First, due to the multi-step nature of decision-making tasks, it would be necessary to execute k different actions corresponding to k rollouts at every interaction step. This would further drive up the cost of producing the corresponding rollout. Second, in stateful environments, the environment transitions to the next state after executing an action. Consequently, obtaining multiple rollouts from the same state would require environment resets, which is not always feasible.
> >
> > Our manuscript has improved substantially by addressing and incorporating your feedback: thank you. We hope to have clarified your remaining questions.
> >
> > [1] Pathak et al., “Curiosity-driven Exploration by Self-supervised Prediction”, 2017\
> > [2] Burda et al., “Exploration by Random Network Distillation”, 2018\
> > [3] Raileanu et al., “RIDE: Rewarding Impact-driven Exploration for Procedurally-generated Environments”, 2020\
> > [4] Zhang et al., “NovelD: A Simple yet Effective Exploration Criterion”, 2021\
> > [5] J. Gao et al., “Navigate the Unknown: Enhancing LLM Reasoning with Intrinsic Motivation Guided Exploration”, 2025.\
> > [6] Shao et al., “DeepSeekMath: Pushing the Limits of Mathematical Reasoning in Open Language Models”, 2024\
> > [7] Ahmadian et al., “Back to Basics: Revisiting REINFORCE Style Optimization for Learning from Human Feedback in LLMs”, 2024

---

> > > ### Comment · Reviewer_HhsF · 2025-11-24
> > > **Official Comment by Reviewer HhsF**
> > >
> > > I thank the authors for their comprehensive rebuttal. The new analyses have effectively addressed my concerns and strengthened the manuscript. Furthermore, a more in-depth investigation into the methodology for extracting the *action/policy distribution* from the LLM/VLM's output would be highly insightful. I fully recognize that this additional work may be beyond the scope of the current paper. However, even including a discussion of these points in an appendix would be a welcome addition to guide future research.

---

> > > > ### Author Response · Authors · 2025-12-03
> > > >
> > > > We thank the reviewer for their response. We are happy that our analyses addressed your remaining concerns and strengthened the manuscript. We agree that adding a more in-depth discussion of our methodology for extracting the action/policy distribution from the LLM output will be valuable for guiding future research. We will include this discussion in our final version.

---

### Official Review · Reviewer_Sto4 · 2025-10-31

**Soundness:** 2
**Presentation:** 2
**Contribution:** 1
**Rating:** 2
**Confidence:** 4

**Summary:**

This paper evaluates LLMs in decision-making scenario, focusing on three failure modes: greediness, frequency bias and the knowing-doing gap. The authors illustrate these failure modes on a multi-armed bandit environment, and further show that RL finetuning can help mitigate them.

The paper is generally well-written and easy to follow. However, the novelty of the work is not clear to me. LLMs have been well studied in decision-making settings, and the failure points studied here (lack of exploration, knowing-doing gap) have been noted in previous works (see weaknesses). Using reinforcement learning finetuning to improve the abilities of LLMs is not novel either. I think this work would be better suited as a workshop contribution.

**Strengths:**

- The multi-armed bandit environment allows to illustrate the three failure modes in a controlled setting

**Weaknesses:**

Major:

- W1: The results presented in this work are not particularly novel.
    1. LLMs have been studied as agents in decision-making environment in numerous works (see for example [1] for a recent review). As the authors themselves note in the introduction, previous works have observed failure modes such as lack of exploration (cf l.41) and the knowing-doing gap (cf l.43-44). The main results of the paper (Section 4.2) illustrate these failure modes on a multi-armed bandit setting but do not provide further insight beyond observing these failure modes in a controlled environment.
    2. The benefits of RL finetuning with verifiable rewards have been established in previous works, for example [2-5]. The authors mention [4-5] saying that in contrast their approach is “specialized for decision-making scenarios” (l.163), even though I’d argue that [5] is precisely focused on decision-making scenarios already.
    3. Results in the Section 4.5 merely illustrate well-studied methods like RL finetuning, behavior cloning/distillation, and test-time scaling.
- W2: The related works do not cover previous works on RL finetuning and LLM-as-an-agent, which makes the novelty compared to previous works unclear.
- W3: In method l.175, “If no valid action is found, a random action is executed”. Weak LLMs might run into invalid actions often, which introduces an unwanted form of epsilon-greedy exploration where epsilon is the rate at which the LLM doesn’t output a valid action. Given the amount of exploration done by the LLM is precisely what is being studied here, it would be preferable to filter out these invalid actions instances instead.
- W4: In Figure 6, results for 20 arms show that RLFT leads to higher regret for the 9B model, but this is not discussed in the text. Also the plots with lower noise in appendix C.2 are missing the 9B-RLFT model.
- W5: In Section 4.2, the conclusion that “this suggest that frequency bias is an artifact of supervised pre-training” (l.326) is misleading: results for the 27B model shows (as the authors note l.324) that a strong model can escape the frequency bias, even though it is trained with supervised pre-training.
- W6: In Section 4.2, the illustration provided Figure 24 suggests that the reason the LLM doesn’t pick previously unseen actions might be due to an imprecision in the prompt phrasing. Does the effect remain when specifying that the model should strictly according to the UCB algorithm? Does the result depend on most actions not having been seen previously like in Figure 24?

Minor:

- l.246: “LLMs model” is redundant
- Why is the result on Tictactoe in the ablation section? This just seems to me like it is looking into the effectiveness of RL finetuning as in Section 4.3.


[1] Cao, Yuji, et al. "Survey on large language model-enhanced reinforcement learning: Concept, taxonomy, and methods." IEEE Transactions on Neural Networks and Learning Systems (2024).

[2] Lambert, Nathan, et al. "Tulu 3: Pushing frontiers in open language model post-training." arXiv preprint arXiv:2411.15124 (2024).

[3] Luong, Trung Quoc, et al. "Reft: Reasoning with reinforced fine-tuning." arXiv preprint arXiv:2401.08967 (2024).

[4] Guo, Daya, et al. "Deepseek-r1: Incentivizing reasoning capability in llms via reinforcement learning." arXiv preprint arXiv:2501.12948 (2025).

[5] Zhai, Simon, et al. "Fine-tuning large vision-language models as decision-making agents via reinforcement learning." Advances in neural information processing systems 37 (2024): 110935-110971.

**Questions:**

- Figure 3: what do the cross markers correspond to?
- Figure 4: the plots (a) and (c) are unclear to me: as I understand the y-axis (action entropy) refers to the entropy of the distribution over predicted actions. But then why do the plots show markers for each individual action? If they correspond to the action that is repeated N times across the x-axis, why would we expect the results to look different when only the action name is changing?

---

> ### Author Response · Authors · 2025-11-21
>
> Thank you for your helpful feedback.
>
>
>
> **Novelty:**
>
> We agree with the reviewer that prior work has identified general issues like poor exploration and that RLFT is an established technique. We outline the differences from previous works in the following.
>
>
>
> 1. Our work follows a great line of research that investigates LLM behavior in simple decision-making scenarios, such as bandits, that focuses specifically on the exploration abilities of LLMs [1,2]. In contrast to existing works, we identify 3 concrete failure modes in such scenarios, quantify them empirically, and subsequently systematically analyze the effects of RL fine-tuning on self-generated CoT rationales on these failure modes. As such, our work provides crucial diagnostic insights for the field. The reviewer is right that the knowing-doing gap has been observed in more complex decision-making scenarios [3,4]. In contrast, we empirically study and quantify this gap, which is enabled by our simple bandit setup.
> 2. We use RLFT not as a novel method, but as a scientific tool to causally probe and mitigate these precisely defined failure modes in simple decision-making scenarios. Our analysis is, to the best of our knowledge, the first to rigorously study how RL feedback on CoT reasoning directly impacts exploration and the outlined biases of LLM agents. The key insight is not that RLFT improves performance, but how it does so by, for example, increasing action coverage (mitigating greediness) and reducing reliance on context statistics (counteracting frequency bias). It is true that [5] also focuses on decision-making scenarios; we rephrased this sentence accordingly. Thank you for pointing this out.
> 3. Regarding the ablations on SFT and applying test-time scaling in Section 4.5, our goal was to use these well-established techniques to understand their specific impacts on the failure modes we identified. For example, our SFT experiments clearly highlight the advantage of leveraging expert data if available. Similarly, our thinking budget ablations aim to highlight the benefits of giving the agent more time to think per decision, while recognizing the difficulties in multi-turn interactions. Therefore, we believe these ablations are valuable additions to our paper and contributions to the scientific literature.
>
> We hope this clarifies the positioning of our paper better.
>
> **Extension of Related Work:**
>
> Thank you for this suggestion. We agree that a comprehensive overview is crucial in this fast-moving field. We appreciate the references provided and, upon review, will integrate them and other relevant literature into an expanded related work section in the appendix to better situate our contributions.
>
> **Invalid actions:**
>
> As correctly noted by the reviewer, we execute a random action if no valid action is found within the generated tokens. While this occurs seldomly, this mechanism is a useful safeguard to ensure robust interaction with the environment. In particular, this safeguard was essential for RLFT to enable continued fine-tuning over many trials. To verify that the safeguard does not bias the experiments conducted in Section 4, we report the fraction of random actions performed across the 50-step horizon for Gemma 2 2/9/27B. Indeed, the safeguard does not introduce any meaningful bias for our results (see Table 2 in Appendix C.5).
>
> **20 arms scenario:**
>
> As mentioned in our Future Work section, the 50-step horizon is not sufficient for the 20-arms scenario. Due to the computational cost of running the experiments, we limited the horizon to 50 steps for all settings. Consequently, the greedy action selection strategies of 9B and 27B ICL outperform the optimal UCB algorithm over this limited horizon. By the shape of the regret curves, it is clear that UCB would exhibit a lower regret over a longer horizon than the LLM agent.
>
> The reviewer is correct that 9B ICL outperforming 9B RLFT in this setting is an interesting observation. One possible hypothesis for this is that the higher short-term regret may be a direct consequence of RLFT, which encourages the agent to explore more. The increased exploration can be costly in the short term but may be essential for long-term performance. Over the limited interaction horizon in the 20-arm scenarios, this may be disadvantageous. We added a paragraph on this point to Appendix C.2.
>
> **Frequency bias with 27B:**
>
> We thank the reviewer for pointing this out. It is indeed true that the 27B model escapes the frequency bias. We corrected the phrasing in our updated manuscript accordingly.

---

> > ### Author Response · Authors · 2025-11-21
> >
> > **Prompt phrasing for Knowing-doing gap:**
> >
> > We understand this valid concern. To ensure that our findings represent a fundamental model bias and not a prompt artifact, we investigated a range of prompting strategies to encourage the LLM to strictly adhere to the UCB algorithm in preliminary experiments. For example, this involved strong language, including explicit instructions to prioritize untried actions or to prefer actions with high/infinite UCB values. However, across prompting strategies, our findings remained similar. Therefore, we do not expect that the bias stems from an imprecision in the prompt phrasing. We added a paragraph on our experience with different prompting strategies to Appendix C.1.3.
> >
> > Regarding your **minor points**:
> >
> > * Thank you for pointing this out. We changed this in our updated manuscript.
> > * This is a valid point. We included the Tic-tac-toe experiment in our ablation section as it provides an initial investigation into the effect of RLFT in stateful environments. Tic-tac-toe served as a core motivation for us to better understand the agentic abilities of LLM agents in decision-making scenarios. In [4], the authors showed that frontier models struggle considerably even when playing Tic-tac-toe against a random agent. In our experiments, we found meaningful positive effects of RLFT in Tic-tac-toe. However, given that we did not deeply study the outlined biases in Tic-tac-toe and for consistency with the general structure of the paper, we kept this in the ablation section.
> >
> > Regarding your additional **questions**:
> >
> > * The cross-markers in Figure 3c indicate the performance of the 2B/9B and 27B models without Chain-of-Though (CoT), while the dots represent the respective performances with CoT. This is indicated by the legend in Figure 3c. We observed that disabling CoT reasoning had a considerable impact on the action coverage and regret.
> > * In Figure 4a/c, we plot the repetition frequency of the target action (x-axis, 0 to 100 repeats) against the action entropy over all possible actions (y-axis). The markers in the plot represent particular target actions (green/red/blue/orange/black buttons). For this experiment, we construct prefix histories (that occurred during environment interaction) and then “synthetically” repeat the last action, the *target action*, 0 to 100 times in the context given to the agent. We ensure that the last action is not the optimal action. For a given context, we then compute the action entropy over all possible next actions. For the 10 arms scenario, every prefix history therefore results in 1000 forward passes (10 arms * 100 repetitions of target action). We repeat this procedure for 5 target actions (Figure 4a/c) using 5 interaction histories per action. This procedure is described in detail in Appendix C.1.2.
> >
> > Thank you again for your comments, which helped us to improve our manuscript considerably. We hope to have clarified the positioning of our paper and your questions. We would appreciate it if you considered raising your score in light of our responses. We are happy to engage in further discussion if questions remain.
> >
> > [1] Krishnamurthy et al., “Can large language models explore in-context?”, 2024\
> > [2] Nie et al., “EVoLVe: Evaluating and Optimizing LLMs for In-context Exploration”, 2025\
> > [3] Paglieri et al., “Balrog: Benchmarking Agentic LLM and VLM Reasoning on Games”, 2025\
> > [4] Ruoss et al., “LMAct: A Benchmark for In-Context Imitation Learning with Long Multimodal Demonstrations”, 2024\
> > [5] Zhai et al., “Fine-Tuning Large Vision-Language Models as Decision-Making Agents via Reinforcement Learning”, 2024

---

### Official Review · Reviewer_ytZh · 2025-11-03

**Soundness:** 3
**Presentation:** 3
**Contribution:** 2
**Rating:** 6
**Confidence:** 4

**Summary:**

This paper empirically investigates why LLMs perform sub-optimally in decision making settings and identifies three primary failure modes across different model families: greediness, frequency bias, and the knowing-doing gap. Through experiments on multi-armed bandits, contextual bandits, and a text-based Tic-tac-toe environment, this work provides an empirical characterization of these issues across model scales (Gemma2 2B/9B/27B, Llama3 3B/8B/70B and Qwen-2.5 3B/7B/14B/32B). The authors have designed a suite of experiments exploring whether Reinforcement Learning Fine Tuning (RLFT) on self-generated Chain-of-Thought (CoT) rationales can help mitigate these issues, showing that RLFT increases exploration, reduces the greediness and frequency bias in decision making and also narrows the knowing-doing gap with an increased reasoning token budget. Additionally, classical and LLM-specific exploration mechanisms (like $\epsilon$-greedy, self-correction, self-consistency) are used to further improve the post-RLFT performance. Ablation studies highlight the roles of CoT traces in RLFT, expert data for behavior cloning vs thought cloning in SFT, and reasoning token budgets for inference time computational efficiency. Overall, this paper offers an insightful analysis into the decision-making behavior of LLMs and the impact of RLFT in addressing some of the current drawbacks of decision making with different language models.

**Strengths:**

- This paper identifies and rigorously quantifies three key failure modes (greediness, frequency bias, knowing-doing gap) in LLMs for decision making applications, specifically focused on bandit-style decision making with instruction-tuned large language models.

- The authors use RLFT to analyze how reinforcement feedback influences reasoning and exploration, with appropriate baselines and ablations across different LLM model families.

- The paper is overall well-written with thorough description of relevant implementation details and baselines, improving reproducibility.

**Weaknesses:**

- RLFT itself is a straightforward application of PPO-style RLHF training to CoT outputs, therefore the methodological novelty lies mainly in the empirical analysis and not in a new algorithm.
- The environments are relatively simple (bandits, Tic-tac-toe) and while appropriate for controlled study, stronger claims about decision-making or agentic abilities would benefit from evaluation on richer, stateful domains.
- The 50-step interaction limit may underestimate the difficulty of exploration and long-term adaptation with LLMs.
- Unclear scalability to larger LLMs: Most analyses are conducted on $\leq$27B models and the implications for larger model scales (e.g. Llama3 70B) are briefly discussed but not thoroughly tested.
- RLFT results seem sensitive to design choices for reward shaping and action space exploration strategies. Ablation experiments in this paper demonstrate the effects qualitatively (in most likely a single trial) but they are not statistically quantified.

**Questions:**

- Beyond empirical results, do authors have any intuitive understanding or insights into why RLFT improves exploration (maybe changes in CoT token distributions or entropy)?

- How would the observations in this work transfer to other recent reinforcement-based fine-tuning frameworks (GRPO, DPO, or self-critique fine-tuning)?

- Do the RLFT improvements transfer across unseen bandit configurations or to different reasoning prompts?

- In both Fig 21 (a) and (b), a larger model does not always outperform a smaller one in terms of action space coverage. Is there any ablation experiment or empirical observation explaining this behavior?

- The current evaluation for the knowing-doing gap is not extended to the Llama3 70B model, which is shown to have less severe issues from greediness and frequency bias. It would help to include empirical evidence of the relationship, if any, between the knowing-doing gap and model size.

---

> ### Author Response · Authors · 2025-11-21
>
> We thank the reviewer for their constructive feedback. We address all your points and questions below.
>
> **Methodological Novelty:**
>
> In this work, we critically study how LLMs behave in simple decision-making scenarios and analyze when and why they may behave suboptimally. In particular, we focus on analysis that aims to provide a foundational understanding of potential LLM failure modes in decision-making, which is a prerequisite for developing future methods. Consequently, our work is not meant to be a method paper, but diagnostic by design. As correctly noted by the reviewer, we are not the first to introduce RL fine-tuning, but rely on established techniques, such as PPO [1]. While we do not propose a new method, we make adaptations to existing approaches to make them suitable for our LLM setting. These changes required careful consideration, and we found them to be critical for effective RLFT (e.g., reward shaping and action extraction, see Section 3.2). We understand that this positioning was not sufficiently clear, and hope that our response and revised manuscript clarify the positioning of our work better.
>
> **Complex environments:**
>
> * In this work, we intentionally focused on simple environments (MABs, tic-tac-toe). This simple setup allows us to study the decision-making abilities of LLMs in a principled way, as correctly identified by the reviewer. This controlled setup allows us to precisely quantify failure modes, such as the knowing-doing gap, which would be difficult to disentangle in more complex stateful environments. However, we agree with the reviewer that studying the effects of RLFT in richer stateful environments in which exploration is critical will be valuable. We have listed this point in our future work section.
> * Even in our restricted setting, we found that LLMs often behave suboptimally, as characterized by the three failure modes that we analyzed. Therefore, we expect that they might exhibit related biases in more complex agentic scenarios, which is also indicated by two recent benchmarks for agentic LLMs [2,3]. For example, in [3], frontier models struggle in more complex stateful environments, such as Crafter. Nevertheless, we did observe positive effects of RLFT on the agentic abilities of LLMs in MABs/Tic-tac-toe, and we expect that RLFT will also help in more complex environments.
>
> **50-step horizon:**
>
> * We limit the interaction horizon to 50 steps for practical reasons. In our setup, the agent has to produce a decision rationale at every step due to the multi-step nature of decision-making. Every step involves both a prefilling stage (encoding instructions, history) and a generation stage for producing the rationale/action. We set the generation budget to 256 tokens by default, and consequently, the agent produces 13K tokens per episode. In particular, for RLFT, this is a considerable computation burden, which we noted in our future work section.
> * For the 5 and 10-arms scenario, we consider the 50-step horizon sufficient to get reliable performance estimates. For the 20-arms scenario, the 50-step horizon can be insufficient (as noted in our Future Work section). However, in Figures 3a and b, we observe a considerable flattening of the action coverages after only 10 steps. This suggests that the agents would not continue to explore even if run for longer.
> * To verify this, we conduct an additional ablation in which we run Gemma 2B and 27B on the 10 arms scenario over an extended horizon of 100 interaction steps (see Figure 18 in our updated manuscript). Indeed, we find that the action coverage remains flat, indicating that the agents suffer from the biases outlined in our manuscript.
>
> **Scalability to larger LLMs:**
>
> We agree with the reviewer that testing larger models beyond 27B parameters or further expanding the analysis with Llama 70B would be valuable. We are unable to perform the additional comparison with Llama 70B within the timeframe of the rebuttal due to the computational cost of running inference and RLFT with larger models. While analyzing larger and frontier models is an important direction, our work puts a deliberate focus on more accessible small-to-medium scale models. We believe that understanding and mitigating the failure models in these widely used models is valuable in its own right. We hope this clarifies the positioning of our work better.
>
> **RLFT sensitivity:**
>
> For all our RLFT experiments, we report the mean and 95% CIs over 3 seeds (see Appendix B.1, Figures 6 and 9). For the exploration mechanism analysis in Figure 8, we report the mean across the three seeds for action coverage and cumulative regret. We opt for this choice to enable a clearer visualization. However, we agree that reporting the intervals is useful for this ablation and aim to add them to the appendix for our final version.

---

> > ### Author Response · Authors · 2025-11-21
> >
> > Regarding your additional **questions**:
> >
> > 1. Intuitively, RLFT aims to maximize the reward. In the environments we consider, it rewards reasoning chains that lead to exploratory actions and reinforces the “thought” process for exploration, rather than just the final action. This is because increasing exploration is the main driver for increasing the reward obtained in our simple environments. For this reason, we assume that RLFT drives exploration abilities. We leave a deeper analysis for future work.
> > 2. The observations we made in this work can inform design decisions of future algorithms in applications that require multi-turn interactions over extended horizons (e.g., in coding, computer use). For example, one important aspect of tool-use agents fine-tuned via GRPO could be a reward-shaping term to diversify tool selection and to counteract potential greediness biases. Similarly, due to the identified biases, it may be necessary to first restrict the number of available tools and only expand them over time (akin to the agentless to SWE-agent approach in Kimi Dev [4]). We added a paragraph on this to our future work section.
> > 3. For RLFT, we maintain a set of training and evaluation bandits. All reported results in the paper are on the evaluation bandits (see Appendix B.2). We sample both training and evaluation bandits from the same distribution (same range for $\mu$, $\sigma$ for Gaussian bandits). We did not study OOD bandit configurations in this work, as we already observed considerable gaps for the in-domain configurations. However, we believe that it would be interesting to study the transfer of the RLFT checkpoints to other environments in follow-up work (e.g., by shifting the means of the arms). In comparison to SFT, RLFT may increase robustness in OOD environments, which would reflect findings by [5]. We added a point on this to our future work section.
> > 4. It is true that for our ablation with Qwen 2.5 in Figure 21a, the larger models do not always outperform the smaller models (e.g., 7B attains 10% higher action coverage than 14B). This is an interesting observation of non-monotonic scaling, where performance on certain tasks does not strictly increase with model size. We believe that this may stem from differences in the pre-training corpora or alignment techniques applied to specific model checkpoints. While a deeper investigation is beyond our scope, we added a note in the paper acknowledging this observation to Appendix C.4. However, across model sizes, it is apparent that the action coverage flattens out over the 50-step horizon.
> > 5. We agree that studying the knowing-doing gap with Llama3 70b in addition to Gemma2 27B would be a valuable addition to the paper. We hypothesize that while the gap may be smaller, it is likely to persist for Llama3 70B, similar to the greediness bias reported in Figure 22. While we cannot add this ablation within the timeframe of the rebuttal, we believe that studying the knowing-doing gap more rigorously in complex environments and with varying models would be fruitful for future work, and added this item to our future work section.
> >
> > Thank you again for your constructive comments, which have substantially helped us to clarify our positioning. If our response clarified your questions, we would appreciate it if you considered raising your score. If questions remain, we would be happy to engage in further discussion.
> >
> > [1] J. Schulman et al., “Proximal Policy Optimization Algorithms”, 2017\
> > [2] Paglieri et al., “Balrog: Benchmarking Agentic LLM and VLM Reasoning on Games”, 2025\
> > [3] Ruoss et al., “LMAct: A Benchmark for In-Context Imitation Learning with Long Multimodal Demonstrations”, 2024\
> > [4] Yang et al., “Kimi-Dev: Agentless Training as Skill Prior for SWE-Agents”, 2025\
> > [5] Chu et al., “SFT Memorizes, RL Generalizes: A Comparative Study of Foundation Model Post-training”, 2025

---

### Meta-Review · Area_Chair_ozD3 · 2026-01-07

**Summary:**

All reviewers acknowledge, and I agree on, the diagnostic value of this paper’s contribution.

**Reviewer Concerns:**

In my view, most of the reviewers’ concerns are well-addressed. The bigger ones are about the less realistic nature of the problem (which allows the kind of systematic study the paper is appreciated for), the practical constraints on the experiment scale, etc. I don’t think the model scale concerns are addressed, but do not believe it is fair to penalize the paper for the cost to run the larger models; these results are valuable.

**Reviewer Scores:**

Reviewer 8Kgn maintains their score, citing “generalizability of these findings to other tasks and larger models.” Reviewer Sto4 did not respond to the rebuttal in time. Though I believe their novelty concerns are well-addressed, it is unclear whether they would be persuaded of this. The remaining two reviews are accept-leaning, and positive enough that I don’t think revising their ratings is necessary to support the paper.

---

### Decision · Program_Chairs · 2026-01-26

Accept (Poster)